

# Variability and quasi-decadal changes in the methane budget over the period 2000-2012

Marielle Saunois[1], Philippe Bousquet[1], Ben Poulter[2], Anna Peregon[1], Philippe Ciais[1], Josep G. Canadell[3], Edward J. Dlugokencky[4], Giuseppe Etiope[5,6], David Bastviken[7], Sander Houweling[8,9], Greet Janssens-Maenhout[10], Francesco N. Tubiello[11], Simona Castaldi[12,13,14], Robert B. Jackson[15], Mihai Alexe[10], Vivek K. Arora[16], David J. Beerling[17], Peter Bergamaschi[10], Donald R. Blake[18], Gordon Brailsford[19], Lori Bruhwiler[4], Cyril Crevoisier[20], Patrick Crill[21], Kristofer Covey[22], Christian Frankenberg[23,24], Nicola Gedney[25], Lena Höglund-Isaksson[26], Misa Ishizawa[27], Akihiko Ito[27], Fortunat Joos[28], Heon-Sook Kim[27], Thomas Kleinen[29], Paul Krummel[30], Jean-François Lamarque[31], Ray Langenfelds[30], Robin Locatelli[1], Toshinobu Machida[27], Shamil Maksyutov[27], Joe R. Melton[32], Isamu Morino[33] Vaishali Naik[34], Simon O'Doherty[35], Frans-Jan W. Parmentier[36], Prabir K. Patra[37], Changhui Peng[38,39], Shushi Peng[1,40], Glen P. Peters[41], Isabelle Pison[1], Ronald Prinn[42], Michel Ramonet[1], William J. Riley[43], Makoto Saito[27], Monia Santini[14], Ronny Schroeder[44], Isobel J. Simpson[18], Renato Spahni[28], Atsushi Takizawa[45], Brett F. Thornton[22], Hanqin Tian[46], Yasunori Tohjima[27], Nicolas Viovy[1], Apostolos Voulgarakis[47], Ray Weiss[48], David J. Wilton[17], Andy Wiltshire[49], Doug Worthy[50], Debra Wunch[51], Xiyan Xu[43,52], Yukio Yoshida[27], Bowen Zhang[46], Zhen Zhang[2,53], and Qiuan Zhu[39].

[1]Laboratoire des Sciences du Climat et de l'Environnement, LSCE-IPSL (CEA-CNRS-UVSQ), Université Paris-Saclay 91191 Gif-sur-Yvette, France
[2]NASA Goddard Space Flight Center, Biospheric Science Laboratory, Greenbelt, MD 20771, USA
[3]Global Carbon Project, CSIRO Oceans and Atmosphere, Canberra, ACT 2601, Australia
[4]NOAA ESRL, 325 Broadway, Boulder, Colorado 80305, USA
[5]Istituto Nazionale di Geofisica e Vulcanologia, Sezione Roma 2, via V. Murata 605 00143 Roma
[6]Faculty of Environmental Science and Engineering, Babes Bolyai University, Cluj-Napoca, Romania.
[7]Department of Thematic Studies – Environmental Change, Linköping University, SE-581 83 Linköping, Sweden
[8]Netherlands Institute for Space Research (SRON), Sorbonnelaan 2, 3584 CA Utrecht, The Netherlands
[9]Institute for Marine and Atmospheric Research Sorbonnelaan 2, 3584 CA, Utrecht, The Netherlands
[10]European Commission Joint Research Centre, Ispra (Va), Italy
[11]Statistics Division, Food and Agriculture Organization of the United Nations (FAO), Viale delle Terme di Caracalla, Rome 00153, Italy
[12]Dipartimento di Scienze Ambientali, Biologiche e Farmaceutiche, Seconda Universita di Napoli, via Vivaldi 43, 81100 Caserta, Italy
[13]Far East Federal University (FEFU), Vladivostok, Russky Island, Russia
[14]Euro-Mediterranean Center on Climate Change, Via Augusto Imperatore 16, 73100 Lecce, Italy
[15]School of Earth, Energy & Environmental Sciences, Stanford University, Stanford, CA 94305-2210, USA
[16]Canadian Centre for Climate Modelling and Analysis, Climate Research Division, Environment and Climate Change Canada, Victoria, BC, V8W 2Y2, Canada
[17]Department of Animal and Plant Sciences, University of Sheffield, Sheffield S10 2TN, UK
[18]University of California Irvine, 570 Rowland Hall, Irvine, California 92697, USA
[19]National Institute of Water and Atmospheric Research, 301 Evans Bay Parade, Wellington, New Zealand
[20]Laboratoire de Météorologie Dynamique, LMD/IPSL, CNRS Ecole polytechnique, Université Paris-Saclay, 91120 Palaiseau, France





[21]Department of Geological Sciences and Bolin Centre for Climate Research, Svante Arrhenius väg 8, SE-106 91 Stockholm, Sweden

[22]School of Forestry and Environmental Studies, Yale University New Haven, CT 06511, USA

[23] California Institute of Technology, Geological and Planetary Sciences, Pasadena, USA

[24]Jet Propulsion Laboratory, M/S 183-601, 4800 Oak Grove Drive, Pasadena, CA 91109, USA

[25]Met Office Hadley Centre, Joint Centre for Hydrometeorological Research, Maclean Building, Wallingford OX10 8BB, UK

[26]Air Quality and Greenhouse Gases program (AIR), International Institute for Applied Systems Analysis (IIASA), A-2361 Laxenburg, Austria

[27]Center for Global Environmental Research, National Institute for Environmental Studies (NIES), Onogawa 16-2, Tsukuba, Ibaraki 305-8506, Japan

[28]Climate and Environmental Physics, Physics Institute and Oeschger Center for Climate Change Research, University of Bern, Sidlerstr. 5, CH-3012 Bern, Switzerland

[29]Max Planck Institute for Meteorology, Bundesstrasse 53, 20146 Hamburg, Germany

[30]CSIRO Oceans and Atmosphere, Aspendale, Victoria 3195 Australia

[31]NCAR, PO Box 3000, Boulder, Colorado 80307-3000, USA

[32]Climate Research Division, Environment and Climate Change Canada, Victoria, BC, V8W 2Y2, Canada

[33]Center for Global Environmental Research, National Institute for Environmental Studies (NIES), Onogawa 16-2, Tsukuba, Ibaraki 305-8506, Japan.

[34]NOAA, GFDL, 201 Forrestal Rd., Princeton, NJ 08540

[35]School of Chemistry, University of Bristol, Cantock's Close, Clifton, Bristol BS8 1TS

[36]Department of Arctic and Marine Biology, Faculty of Biosciences, Fisheries and Economics, UiT: The Arctic University of Norway, NO-9037, Tromsø, Norway

[37]Department of Environmental Geochemical Cycle Research and Institute of Arctic Climate and Environment Research, JAMSTEC, 3173-25 Showa-machi, Kanazawa-ku, Yokohama, 236-0001, Japan

[38]Department of Biology Sciences, Institute of Environment Science, University of Quebec at Montreal, Montreal, QC H3C 3P8, Canada

[39]State Key Laboratory of Soil Erosion and Dryland Farming on the Loess Plateau, Northwest A&F University, Yangling, Shaanxi 712100, China

[40]Sino-French Institute for Earth System Science, College of Urban and Environmental Sciences, Peking University, Beijing 100871, China

[41]CICERO Center for International Climate Research, Pb. 1129 Blindern, 0318 Oslo, Norway

[42]Massachusetts Institute of Technology (MIT), Building 54-1312, Cambridge, MA 02139, USA

[43]Earth Sciences Division, Lawrence Berkeley National Lab, 1 Cyclotron Road, Berkeley, CA 94720, USA

[44]Department of Civil & Environmental Engineering, University of New Hampshire, Durham, NH 03824, USA

[45]Japan Meteorological Agency (JMA), 1-3-4 Otemachi, Chiyoda-ku, Tokyo 100-8122, Japan

[46]International Center for Climate and Global Change Research, School of Forestry and Wildlife Sciences, Auburn University, 602 Duncan Drive, Auburn, AL 36849, USA

[47]Space & Atmospheric Physics, The Blackett Laboratory, Imperial College London, London SW7 2AZ, U.K.

[48]Scripps Institution of Oceanography (SIO), University of California San Diego, La Jolla, CA 92093, USA

[49]Met Office Hadley Centre, FitzRoy Road, Exeter, EX1 3PB, United Kingdom

[50]Environnement Canada, 4905, rue Dufferin, Toronto, Canada.

[51]Department of Physics, University of Toronto, 60 St. George Street, Toronto, Ontario, Canada

[52]CAS Key Laboratory of Regional Climate-Environment for Temperate East Asia, Institute of Atmospheric Physics, Chinese Academy of Sciences, Beijing 100029, China

[53]Swiss Federal Research Institute WSL, Birmensdorf 8059, Switzerland

*Correspondance to*: Marielle Saunois (marielle.saunois@lsce.ipsl.fr)





**Abstract.** Following the recent Global Carbon project (GCP) synthesis of the decadal methane ($CH_4$) budget over 2000-2012 (Saunois et al., 2016), we analyse here the same dataset with a focus on quasi-decadal and inter-annual variability in $CH_4$ emissions. The GCP dataset integrates results from top-down studies (exploiting atmospheric observations within an

atmospheric inverse-modelling frameworks) and bottom-up models, inventories, and data-driven approaches (including process-based models for estimating land surface emissions and atmospheric chemistry, inventories of anthropogenic emissions, and data-driven extrapolations).

The annual global methane emissions from top-down studies, which by construction match the observed methane growth rate within their uncertainties, all show an increase in total methane emissions over the period 2000-2012, but this increase is

not linear over the 13 years. Despite differences between individual studies, the mean emission anomaly of the top-down ensemble shows no significant trend in total methane emissions over the period 2000-2006, during the plateau of atmospheric methane mole fractions, and also over the period 2008-2012, during the renewed atmospheric methane increase. However, the top-down ensemble mean produces an emission shift between 2006 and 2008, leading to 22 [16-32] Tg $CH_4$ $yr^{-1}$ higher methane emissions over the period 2008-2012 compared to 2002-2006. This emission increase mostly originated

from the tropics with a smaller contribution from mid-latitudes and no significant change from boreal regions.

The regional contributions remain uncertain in top-down studies. Tropical South America and South and East Asia seems to contribute the most to the emission increase in the tropics. However, these two regions have only limited atmospheric measurements and remain therefore poorly constrained.

The sectorial partitioning of this emission increase between the periods 2002-2006 and 2008-2012 differs from one

atmospheric inversion study to another. However, all top-down studies suggest smaller changes in fossil fuel emissions (from oil, gas, and coal industries) compared to the mean of the bottom-up inventories included in this study. This difference is partly driven by a smaller emission change in China from the top-down studies compared to the estimate in the EDGARv4.2 inventory, which should be revised to smaller values in a near future. Though the sectorial partitioning of six individual top-down studies out of eight are not consistent with the observed change in atmospheric $^{13}CH_4$, the partitioning

derived from the ensemble mean is consistent with this isotopic constraint. At the global scale, the top-down ensemble mean suggests that, the dominant contribution to the resumed atmospheric $CH_4$ growth after 2006 comes from microbial sources (more from agriculture and waste sectors than from natural wetlands), with an uncertain but smaller contribution from fossil $CH_4$ emissions. Besides, a decrease in biomass burning emissions (in agreement with the biomass burning emission databases) makes the balance of sources consistent with atmospheric $^{13}CH_4$ observations.

The methane loss (in particular through OH oxidation) has not been investigated in detail in this study, although it may play a significant role in the recent atmospheric methane changes.





# 1 Introduction

Methane (CH$_4$), the second most important anthropogenic greenhouse gas in terms of radiative forcing, is highly relevant to mitigation policy due to its shorter lifetime and its stronger warming potential compared to carbon dioxide. Atmospheric CH$_4$ mole fraction has experienced a renewed and sustained increase since 2007 after almost ten years of stagnation
(Dlugokencky et al., 2009; Rigby et al., 2008; Nisbet et al., 2014, 2016). Over 2006-2013, the atmospheric CH$_4$ growth rate was about 5 ppb yr$^{-1}$, before reaching 12.7 ppb yr$^{-1}$ in 2014 and 9.5 ppb yr$^{-1}$ in 2015 (NOAA monitoring network: http://www.esrl.noaa.gov/gmd/ccgg/trends_ch4/).

The growth rate of atmospheric methane is a very accurate measurement of the imbalance between global sources and sinks. Methane is emitted by anthropogenic sources (livestock including enteric fermentation and manure management; rice
cultivation; solid waste and wastewater; fossil fuel production, transmission and distribution; biomass burning), and natural sources (wetlands; and other inland freshwater, geological sources, hydrates, termites, wild animals). Methane is mostly destroyed in the atmosphere by hydroxyl radical (OH) oxidation (90 % of the atmospheric sink). Other sinks include destruction by atomic oxygen and chlorine in the stratosphere and in the marine boundary layer for the latter, and upland soil sink by microbial methane oxidation. The changes in these sources and sinks can be investigated by different methods:
bottom-up process-based models of wetland emissions (Melton et al., 2013; Bohn et al., 2015; Poulter et al., 2016), rice paddy emissions (Zhang et al. 2016), termite emissions (Sanderson, 1996; Kirschke et al., 2013, supplementary) and soil uptake (Curry, 2007), data-driven approaches for other natural fluxes (e.g., Bastviken et al. (2011); Etiope (2015)), atmospheric chemistry climate model for methane oxidation by OH (John et al. 2012; Naik et al., 2013; Voulgarakis et al., 2013), bottom-up inventories for anthropogenic emissions (e.g., EDGAR, EPA, FAO, GAINS), observation-driven models
for biomass burning emissions (e.g., GFED) and finally by atmospheric inversions, which optimally combine methane atmospheric observations within a chemistry transport model, and a prior knowledge of sources and sinks (inversions are also called top-down approaches, e.g., Bergamaschi et al. (2013); Houweling et al. (2014); Pison et al. (2013)).

The renewed increase in atmospheric methane since 2007 has been investigated in the recent past years; atmospheric concentration-based studies suggest a mostly tropical signal, with a small contribution from the mid-latitudes and no clear
change from high latitudes (Bousquet et al., 2011; Bergamaschi et al., 2013; Bruhwiler et al., 2014; Dlugokencky et al., 2011; Patra et al., 2016; Nisbet et al., 2016). The year 2007 was found to be a year with exceptionally high emissions from the Arctic (e.g., Dlugokencky et al. (2009)), but it does not mean that Arctic emissions were persistently higher during the entire period 2008-2012. Attribution of the renewed atmospheric CH$_4$ growth to specific source and sink processes is still being debated. Bergamaschi et al. (2013) found that anthropogenic emissions were the most important contributor to the
methane growth rate increase after 2007, though smaller than in the EDGARv4.2FT2010 inventory. In contrast, Bousquet et al. (2011) explained the methane increases in 2007-2008 by an increase mainly in natural emissions, while Poulter et al. (in review) do not find significant trends in global wetland emissions from an ensemble of wetland models over the period 2000-2012. McNorton et al. (2016b) using a single wetland emission model with a different wetland dynamics scheme also





concluded a small increase (3%) in wetland emissions relative to 1993-2006. Associated with the atmospheric $CH_4$ mixing ratio increase, the atmospheric $\delta^{13}C\text{-}CH_4$ shows a continuous decrease since 2007 (e.g., Nisbet al. (2016)), pointing towards increasing sources with depleted $\delta^{13}C\text{-}CH_4$ (microbial) and/or decreasing sources with enriched $\delta^{13}C\text{-}CH_4$ (pyrogenic, thermogenic). Using a box model combining $\delta^{13}C\text{-}CH_4$ and $CH_4$ observations, two recent studies infer a dominant role of

increasing microbial emissions (more depleted in $^{13}C$ than thermogenic and pyrogenic sources) to explain the higher $CH_4$ growth rate after circa. 2006. Schaefer et al. (2016) hypothesized (but did not prove) that the increasing microbial source was from agriculture rather than from natural wetlands, however given the uncertainties in isotopic signatures the evidence against wetlands is not strong. Schwietzke et al. (2016), using different estimates of the source isotopic signatures with rather narrow uncertainty ranges, also find a positive trend in microbial emissions. In a scenario where biomass burning emissions

are constant over time, they inferred decreasing fossil fuel emissions, in disagreement with emission inventories. However, the global burned area is suggested to have decreased (-1.2% yr$^{-1}$) over the period 2000-2012 (Giglio et al., 2013) leading to a decrease in biomass burning emissions ([http://www.globalfiredata.org/figures.html](http://www.globalfiredata.org/figures.html)). In a second scenario including a 1.2 % yr$^{-1}$ decrease in biomass burning emissions, Schwietzke et al. (2016) find fossil fuel emissions close to constant over time, when coal production significantly increased, mainly from China. Atmospheric observations of ethane, a species co-emitted

with methane in the oil and gas up-stream sector can be used to estimate methane emissions from this sector (e.g., Wennberg et al. (2012)). Using such a method, Hausmann et al. (2016) suggested a significant increase in oil and gas methane emissions contributing to the increase in total methane emissions. However, these studies rely on emission ratios of ethane to methane, which are uncertain and may vary substantially over the years (e.g., Wunch et al. (2016)); yet this potential variation over time is not well documented. The increase in methane mole fractions could also be due to a decrease in OH

global concentrations (Rigby et al., 2008; Holmes et al., 2013). Although OH year-to-year variability appears to be smaller than previously thought (e.g., Montzka et al. (2011)), a long-term trend can still strongly impact the atmospheric methane growth rate as a 1% change in OH corresponds to a 5 Tg change in methane emissions (Dalsoren et al., 2009). Indeed, after an increase in OH concentrations over the period 1970-2007, Dalsoren et al. (2016) found constant OH concentration since 2007, which could contribute to the observed increase in methane growth rate and therefore limit the required changes in

methane emissions inferred by top-down studies.

Using top-down approaches, an accurate attribution of changes in methane emissions per region is difficult due to the sparse coverage of surface networks (e.g., Dlugokencky et al. (2011)). Satellite data offer a better coverage in some poorly sampled regions (tropics), and progress has been made in improving satellite retrievals of $CH_4$ column mole fractions (e.g., Butz et al. (2011); Cressot et al. (2014)). Yet the complete exploitation of remote sensing of $CH_4$ column gradients in the atmosphere to

infer regional sources is still limited by relatively poor accuracy and gaps in the data, although progress has been made moving from SCIAMACHY to GOSAT (Buchwitz et al., 2015; Cressot et al., 2016). Also the chemistry transport models often fail to reproduce correctly the methane vertical gradient, especially in the stratosphere (Saad et al., 2016; Wang et al., 2016) and this misrepresentation in the models may impact the inferred surface fluxes when constrained by total column



observations. Furthermore, uncertainties in top-down estimates stem from uncertainties in atmospheric transport and the setup and data used in the inverse systems (Locatelli et al., 2015; Patra et al., 2011).

One approach to address inversion uncertainties is to gather an ensemble of transport models and inversions. Instead of interpreting one single model to discuss the methane budget changes, here we take advantage of an ensemble of published

studies to extract robust changes and patterns observed since 2000 and in particular since the renewed increase after 2007. This approach allows accounting for the model-to-model uncertainties in detecting robust changes of emissions (Cressot et al., 2016). Attributing sources to sectors (e.g. agriculture vs. fossil) or types (e.g. microbial vs. thermogenic) using inverse systems is challenging if no additional constraints, such as isotopes, are used to separate the different methane sources, which often overlap geographically. Assimilating only $CH_4$ observations, the separation of different sources relies only on

their different seasonality (e.g., rice cultivation, biomass burning, wetlands), on the signal of synoptic peaks related to regional emissions when continuous observations are available, or on distinct spatial distributions. Using isotopic information such as $\delta^{13}C$-$CH_4$ brings some additional constraints on source partitioning to separate microbial vs. fossil and fire emissions, but $\delta^{13}C$-$CH_4$ alone cannot further separate microbial emissions between agriculture, wetlands, termites or freshwaters with enough confidence due to uncertainties in their close isotopic signatures.

The Global Carbon Project (GCP) has provided a collaborative platform for scientists from different disciplinary fields to share their individual expertise and synthesize the current understanding of the global methane budget. Following the first global methane budget published by Kirschke et al. (2013) and using the same dataset as the budget update by Saunois et al. (2016) for 2000-2012, we analyse here the results of an ensemble of top-down and bottom-up approaches in order to determine the robust features that could explain the variability, and quasi-decadal changes in $CH_4$ growth rate since 2000. In

particular, this paper aims to highlight the most likely emission changes that could contribute to the observed positive trend in methane mole fractions since 2007. However, we do not address in detail the contribution of OH changes during this period, as most of the inversions used here assume constant OH concentrations over years, generally only optimizing its mean global concentration against methyl chloroform observations (e.g. Montzka et al. (2011)). It should be kept in mind that any OH change in the atmosphere will limit (in case of decreasing OH) or enhance (in case of increasing OH) the

methane emission changes that are required to explain the observed atmospheric methane recent increase (e.g., Dalsoren et al. (2016)), as further discussed in Sect. 4.

Section 2 presents the ensemble of bottom-up and top-down approaches used in this study as well as the common data processing operated. The main results based on this ensemble are presented and discussed in Sect. 3 through global and regional assessments of the methane emission changes as well as process contributions. We discuss these results in Sect. 4 in

the context of the recent literature summarized in the introduction, and draw some conclusions in Sect. 5.





### 2 Methods

The datasets used in this paper were those collected and published in *The Global Methane Budget 2000-2012* (Saunois et al., 2016). The decadal budget is publicly available at http://doi.org/10.3334/CDIAC/Global_Methane_Budget_2016_V1.1 and on the Global Carbon Project website. Here, we only describe the main characteristics of the data sets and the reader may

refer to the aforementioned detailed paper. The datasets include an ensemble of global top-down approaches as well as bottom-up estimates of the sources and sinks of methane.

**Top-down studies.** The top-down estimates of methane sources and sinks are provided by eight global inverse systems, which optimally combine a prior knowledge of fluxes with atmospheric observations, both with their associated uncertainties, into a chemistry transport model in order to infer methane sources and sinks at specific spatial and temporal

scales. Eight inverse systems have provided a total of 30 inversions over 2000-2012 or shorter periods (Table 1). The longest time series of optimized methane fluxes are provided by inversions using surface in-situ measurements (15). Some surface based inversions were provided over time periods shorter than 10 years (7). Satellite-based inversions (8) provide estimates over shorter time periods (2003-2012 with SCIAMACHY; from June 2009 to 2012 using TANSO/GOSAT). As a result, the discussion presented in this paper will be essentially based on surface-based inversions as GOSAT offers too short a time

series and SCIAMACHY is associated with large systematic errors that need ad-hoc corrections (e.g., Bergamaschi et al. (2013)). Most of the inverse systems estimate the total net methane emission fluxes at the surface (i.e., surface sources minus soil sinks), although some systems solve for a few individual source categories (Table 1). In order to speak in terms of emissions, each inversion provided its associated soil sink fluxes that have been added to the associated net methane fluxes to obtain estimates of surface sources. Saunois et al. (2016) attempted to separate top-down emissions into five categories:

wetland emissions, other natural emissions, emissions from agriculture and waste handling, biomass burning emissions (including agricultural fires), and fossil fuel related emissions. To obtain these individual estimates from those inversions only solving for the net flux, the prior contribution of each source category was used to split the posterior total sources into individual contributions.

**Bottom-up studies.** The bottom-up approaches gather inventories for anthropogenic emissions (agriculture and waste

handling, fossil fuel related emissions, biomass burning emissions), land surface models (wetland emissions), and diverse data-driven approaches (e.g, local measurement up-scaling) for emissions from fresh waters and geological sources (Table 2). Anthropogenic emissions are from the Emissions Database for Global Atmospheric Research (EDGARv4.1, 2010; EDGARV4.2FT2010, 2013), the United States Environmental Protection Agency, USEPA (USEPA, 2006; 2012) and the Greenhouse gas and Air pollutant Interactions and Synergies (GAINS) model developed by the International Institute for

Applied Systems Analysis (IIASA) (Höglund-Isaksson, 2012). They report methane emissions from the following major sources: livestock (enteric fermentation and manure management); rice cultivation; solid waste and wastewater; fossil fuel production, transmission, and distribution. However, they differ in the level of detail by sector and by country, and by the emission factors used for some specific sectors and countries (Höglund-Isaksson et al., 2015). The Food and Agriculture





Organization (FAO) FAOSTAT Emissions dataset (FAOSTAT, 2017a,b) contains estimates of agricultural and biomass burning emissions (Tubiello et al., 2013; 2015). Biomass burning emissions are also taken from the Global Fire Emission Database (version GFED3, van der Werf et al. (2010) and version GFED4s (Giglio et al., 2013; Randerson et al., 2012)), the Fire Inventory from NCAR (FINN, Wiedinmyer et al., (2011)), and the Global Fire Assimilation System (GFAS, Kaiser et

al., (2012)). For wetlands, we use the results of eleven land surface models driven by the same dynamic flooded area extent dataset from remote sensing (Schroeder et al., 2015) over the 2000-2012 period. These models differ mainly in their parameterizations of $CH_4$ flux per unit area in response to climate and biotic factors (Poulter et al., in review; Saunois et al., 2016).

**Data analysis.** The top-down and bottom-up estimates are gathered separately and compared as two ensembles for

anthropogenic, biomass burning, and wetland emissions. For the bottom-up approaches, the category called "other natural" encompasses emissions from termites, wild animals, lakes, oceans, and natural geological seepage (Saunois et al., 2016). However for most of these sources, limited information is available regarding their spatiotemporal distributions. Most of the inversions used here include termite and ocean emissions in their prior fluxes; some also include geological emissions (Table S1). However the emission distributions used by the inversions as prior fluxes are climatological and do not include any inter

annual variability. Geological methane emissions have played a role in past climate changes (Etiope et al., 2008). There is no study on decadal changes in geological $CH_4$ emissions on continental and global scale, although it is known that they may increase or decrease in relation to seismic activity and variations of groundwater hydrostatic pressure (i.e. aquifer depletion). Ocean emissions have been revised downward recently (Saunois et al., 2016). Inter decadal changes in lake fluxes cannot be made in reliable ways because due to the data scarcity and lack of validated models (Saunois et al. 2016). As a result of a

lack of quantified evidences, variations of lakes, oceans, and geological sources are ignored in our bottom-up analysis. However, it should be noted that possible variations of these sources are accounted for in the top-down approaches in the "other natural" category.

Some results are presented as box plots showing the 25%, 50%, and 75% percentiles. The whiskers show minimum and maximum values excluding outliers, which are shown as stars. The mean values are plotted as "+" symbols on the box plot.

The values reported in the text are the mean (XX), minimum (YY) and maximum (ZZ) values as XX [YY-ZZ]. Some estimates rely on few studies so that meaningful 1-sigma values cannot be computed. To consider that methane changes are positive or negative for a time-period (e.g., Fig. 3 and 4 in Sect. 3), we consider that the change is robustly positive or negative when both the first and third quartiles are positive or negative, respectively.

## 3 Results

### 3.1 Global methane variations in 2000-2012

**Atmospheric changes.** The global average methane mole fractions are from four in-situ atmospheric observation networks: the Earth System Research Laboratory from the US National Oceanic and Atmospheric Administration (ESRL-NOAA,




Dlugokencky et al., 1994), the Advanced Global Atmospheric Gases Experiment (AGAGE, Rigby et al., 2008), the Commonwealth Scientific and Industrial Research Organisation (CSIRO, Francey et al., 1999) and the University of California (UCI, Simpson et al. (2012)). The four networks show a consistent evolution of the globally averaged methane mole fractions (Fig. 1a). The methane mole fractions refer here to the same NOAA2004A $CH_4$ reference scale. The different

sampling sites used to compute the global average and the sampling frequency may explain the observed differences between networks. Indeed, the UCI network samples atmospheric methane in the Pacific Ocean between 71°N to 47°S using flasks during specific campaign periods while other networks use both continuous and flask measurements worldwide. During the first half of the 2000s, methane mole fraction remained relatively stable (1770-1785 ppb) with small positive growth rate until 2007 ($0.6\pm0.1$ ppb yr$^{-1}$, Fig. 1b). Since 2007, methane atmospheric mole fraction rose again reaching 1820

ppb in 2012. A mean growth rate of $5.2\pm0.2$ ppb yr$^{-1}$ over the period 2008-2012 is observed (Fig. 1b).

**Global emission changes in individual inversions.** As found in several studies (e.g., Bousquet et al. (2006)), the flux anomaly (see Supplementary, Sect. 2) from top-down inversions (Fig. 1d) is found more robust than the total source estimate when comparing different inversions (Fig. 1c). The mean range between the inverse estimates of total global emissions (Fig. 1c) is of 35 Tg $CH_4$ yr$^{-1}$ (14 to 54 over the years and inversions reported here); this means that the uncertainty in the total

annual global methane emissions inferred by top-down approaches is about 6% (35 Tg $CH_4$ yr$^{-1}$ over 550 Tg $CH_4$ yr$^{-1}$). The three top-down studies spanning 2000 to 2012 (Table 1) show an increase of 15 to 33 Tg $CH_4$ yr$^{-1}$ between 2000 and 2012 (Fig. 1d). Despite the increase in global methane emissions being of the order of magnitude of the range between the models, flux anomalies clearly shows that all individual inversions infer an increase in methane emissions over the period 2000-2012 (Fig. 1d). The inversions using satellite observations included here mainly use GOSAT retrievals (starting from mid-2009)

and only one inversion is constrained with SCIAMACHY column methane mole fractions (from 2003 but ending in 2012, dashed lines in Fig. 1d). On average, satellite-based inversions infer higher annual emissions than surface-based inversions (+12 Tg $CH_4$ yr$^{-1}$ higher over 2010-2012) as previously shown in Saunois et al. (2016) and Locatelli et al. (2015). Also it is worth noting that the ensemble of top-down results shows emissions that are consistently lower in 2009 and higher in 2008 and 2010 (Fig. 1c and Fig. S1).

**Year-to-year changes.** When averaging the anomalies in global emissions over the inversions, we find a difference of 22 [5-37] Tg $CH_4$ between the yearly averages for 2000 and 2012 (Fig. 2a). Over the period 2000-2012, the variations in emission anomalies reveal both year-to-year changes and a positive long-term trend. Year-to-year changes are found to be the largest in the tropics: up to +/- 15 Tg $CH_4$ yr$^{-1}$ (Fig. 2b), with a negative anomaly in 2004-2006 and a positive anomaly after 2007 visible in all inversions except one (Fig. 1d). Compared with the tropical signal, mid-latitude emissions exhibit smaller

anomalies (mean anomaly mostly below 5 Tg $CH_4$ yr$^{-1}$, except around 2005) but contribute a rather sharp increase in 2006-2008 marking a transition between the period 2002-2006 and the period 2008-2012 at the global scale (Fig. 2a and 2c). The boreal regions do not contribute significantly to year-to-year changes, except in 2007, as already noted in several studies (Dlugokencky et al., 2009; Bousquet et al., 2011).





When splitting global methane emissions into anthropogenic and natural emissions at the global scale (Fig. 2e and 2f, respectively), both of these two general categories show significant year-to-year changes. As natural and anthropogenic emissions occur concurrently in several regions; top-down approaches have difficulty in separating their contribution. Therefore the year-to-year variability allocated to anthropogenic emissions from inversions may be an artefact of our

separation method (see Sect. 2) and/or reflect the larger variability between studies compared to natural emissions. However, some of the anthropogenic methane sources are sensitive to climate, such as rice cultivation or biomass burning, and also, to a lesser extent, enteric fermentation and waste management. Fossil-fuel exploitation can also be sensitive to rapid economic changes. However, anthropogenic emissions reported by bottom-studies (black line on Fig. 2e) show much less year-to-year changes then inferred by top-down inversions (blue line of Fig. 2e). China coal production rose faster from 2002 until 2011

when its production started to stabilize or even decline (IEA, 2016). The global natural gas global production steadily increased over time despite a short drop in production in 2009 following the economic crisis (IEA, 2016). The bottom-up inventories do reflect some of this variation such as in 2009 when gas and oil methane emissions slightly decreased (EDGARv4.2FT2010 and EDGARv4.2EXT, Fig. S7). Methane emissions from agriculture and waste are continuously growing in the bottom-up inventories at the global scale. The observed activity data underlying the emissions from

agriculture estimated in this study, as reported by countries to FAO via the FAOSTAT database (FAO, 2017 a,b), exhibit inter annual variabilities that partly explain the variability in methane emissions discussed herein. Livestock methane emissions from America (mainly South America) increased mainly between 2000 and 2004, and remained stable afterwards (estimated by FAOSTAT, Fig. S12). Asian (India, China and, South and East Asia) livestock emissions mainly increased between 2004 and 2008, and remained also rather stable afterwards. On the contrary, livestock emissions in Africa increased

continuously over the full period. These continental variations translate into global livestock emissions increasing continuously over the full period, though at slower rate after 2008 (Fig. S13). Overall, these anthropogenic emissions exhibit more semi-decadal to decadal evolutions (see below) than year-to-year changes as found in top-down inversions.

For natural sources, the mean anomaly of the top-down ensemble suggests year-to-year changes ranging $\pm$ 10 Tg CH$_4$ yr$^{-1}$, lower than but in phase with the total source mean anomaly. The mean anomaly of global natural sources inferred by top-

down studies is negative around 2005 and positive around 2007 (Fig. 2f). The year-to-year variation in wetland emissions inferred from land surface models is of the same order of magnitude but out of phase compared to the ensemble mean top-down estimates (Fig. 2f). However, some individual top-down approaches suggest anomalies smaller than or of different sign to the mean of the ensemble (Fig. S2). Also some land surface models show anomalies in better agreement with the top-down ensemble mean in 2000-2006 (Fig. S11). The 2009 (2010) negative (positive) anomaly in wetland emissions is

common to all land surface models (Fig. S11), and is the result of variations in flooded areas (mainly in the Tropics) and temperature (mainly in boreal regions) (Poulter et al., in review). Overall, from the contradictory results from top-down and bottom-up approaches it is difficult to draw any robust conclusions on the year-to-year variations in natural methane emissions over the period 2000-2012.




**Decadal trend**. The mean anomaly of the inversion estimates shows a positive linear trend in global emissions of $+2.2 \pm 0.2$ Tg $CH_4$ $yr^{-2}$ over 2000-2012 Fig. 2a). It originates mainly from increasing tropical emissions ($+1.6 \pm 0.1$ Tg $CH_4$ $yr^{-2}$, Fig. 2b) with a smaller contribution from the mid-latitudes ($+0.6 \pm 0.1$ Tg $CH_4$ $yr^{-2}$, Fig. 2c). The positive global trend is explained mostly by an increase in anthropogenic emissions, as separated in inversions ($+2.0 \pm 0.1$ Tg $CH_4$ $yr^{-2}$, Fig. 2e).

This represents an increase of about 26 Tg $CH_4$ in the annual anthropogenic emissions between 2000 and 2012, casting serious doubt on the bottom-up methane inventories for anthropogenic emissions, showing an increase in anthropogenic emissions of $+55$ [45-73] Tg $CH_4$ between 2000 and 2012, with USEPA and GAINS inventories at the lower end and EDGARv4.2FT2012 at the higher end of the range. This possible overestimation of the recent anthropogenic emissions increase by inventories has already been suggested in individual studies (e.g., Patra et al. (2011); Bergamaschi et al. (2013);

Bruhwiler et al. (2014); Thompson et al. (2015); Peng, et al. (2016); Saunois et al. (2016)) and is confirmed in this study as a robust feature. Splitting the anthropogenic sources into the components identified in the method section, the trend in anthropogenic emissions from top-down studies mainly originates from the agriculture and waste sector ($+1.2 \pm 0.1$ Tg $CH_4$ $yr^{-2}$, Fig. 3a). Adding the fossil fuel emission trend almost matches the global trend of anthropogenic emissions (Fig. 3b). It should be noted here that the individual inversions all suggest constant to increasing emissions from agriculture and waste

handling (Fig. S3), while some suggest constant to decreasing emissions from fossil fuel use and production (Fig. S4). The latter result seems surprising in view of large increases in coal production during 2000-2012, especially in China. The trend in biomass burning emissions is small but barely significant between 2000 and 2012 ($-0.05 \pm 0.05$ Tg $CH_4$ $yr^{-2}$, Fig. 3). This result is consistent with the GFED dataset (both versions 3 and 4s) for which no significant trend was found over this 13-year period. However, between 2002 and 2010, a significant negative trend of $-0.5 \pm 0.1$ Tg $CH_4$ $yr^{-2}$ is found for biomass

burning, both from the top-down approaches (Fig. S5) and the GFED3 and GFED4s inventory (Fig. S10), though it should be noted that almost all inversions use GFED3 in their prior (Table S1) and therefore are not independent. Over the 13-year period, the wetland emissions in the inversions show a small positive trend ($+0.2 \pm 0.1$ Tg $CH_4$ $yr^{-2}$) about twice the trends of emissions from land surface models but within the range of uncertainty ($+0.1 \pm 0.1$ Tg $CH_4$ $yr^{-2}$, Poulter et al., in review). As stated previously, the wetland emissions from some land surface models disagree with the ensemble mean of land surface

models (Fig. S11).

**Quasi-decadal changes in the period 2000-2012**. According to Fig. 2a, the period 2000-2012 is split into two parts, before 2006 and after 2008. Neither a significant nor a systematic trend in the global total sources (among the inversions of Fig. 1d) is observed before 2006, likewise after 2008 (see Fig. S6 for individual calculated trends); although large year-to-year variations are visible. Before 2006, anthropogenic emissions show a positive trend of $+2.4 \pm 0.2$ Tg $CH_4$ $yr^{-2}$, compensated

by decreasing natural emissions ($-2.4 \pm 0.2$ Tg $CH_4$ $yr^{-2}$) (calculated from Fig. 2e and 2f), which explains the rather stable global total emissions. Bousquet et al. (2006) discussed such compensation between 1999 and 2003. The behaviour of the top-down ensemble mean is consistent with a decrease in microbial emissions in 2000-2006, especially in the northern hemisphere as suggested by Kai et al. (2011) using $^{13}CH_4$ observations. However, some individual top-down studies still suggest constant emissions from both natural and anthropogenic sources (Fig. S2, S3 and S4) over that period as found by





Levin et al. (2012) or Schwietzke et al. (2016), both using also $^{13}CH_4$ observations. The different trends in anthropogenic and natural methane emissions among the inversions highlight the difficulties of the top-down approach to separate natural from anthropogenic emissions and also its dependence on prior emissions. All inversions are based on EGDAR inventory (most of them using EDGARv4.2 version, Table S1). However, the inversions based on the same prior wetland fluxes do not

systematically infer the same variations in methane total and natural emissions, illustrating the freedom of the inversions to deviate from their prior. Contrary to the ensemble mean of inversions, the land surface models gathered in this study show on average a small positive trend ($+0.7 \pm 0.1$ Tg $CH_4$ yr$^{-2}$) during 2000-2006 (calculated from Fig. 2f), with some exceptions in individuals models (Fig. S11). Recently, Schaefer et al. (2016), based on isotopic data, suggested that diminishing thermogenic emissions caused the early 2000s plateau, without ruling out variations in the OH sink. However another

scenario explaining the plateau could combine both constant total sources and sinks. Over 2000-2006, no decrease in thermogenic emissions is found in any of the inversions included in our study (Fig. S4). Even using time-constant prior emissions for fossil fuels in the inversions leads to robustly infer increasing fossil fuel emissions after 2000, although less than when using inter-annually varying prior estimates from inventories (e.g., Bergamaschi et al. (2013)).

All inversions show increasing emissions in the second half of the period, after 2006. For the period 2006-2012, most

inversions show a significant positive trend (below 5 Tg $CH_4$ yr$^{-2}$), within 2-sigma uncertainty for most of the available inversions (see Fig S6). Most of this positive trend is explained by the years 2006 and 2007, due to both natural and anthropogenic emissions, but appears to be highly sensitive to the period of estimation (Fig S6). Between 2008 and 2012, neither the total anthropogenic nor the total natural sources present a significant trend leading to rather stable global total methane emissions (Fig. 2e and 2f). Overall, these results suggest that emissions shifted between 2006 and 2008, rather than

continuously increasing emissions after 2006. Because of this, in the following section, we analyse in more details the emission changes between two time periods: 2002-2006 and 2008-2012 at global and regional scales.

### 3.2 The methane emission changes between 2002-2006 and 2008-2012

### 3.2.1 Global and hemispheric changes inferred by top-down inversions

Integrating all inversions covering at least three years over each 5-year period, the global methane emissions are estimated at

545 [530-563] Tg $CH_4$ yr$^{-1}$ on average over 2002-2006 and at 569 [546-581] Tg $CH_4$ yr$^{-1}$ over 2008-2012. It is worth noting some inversions do not contribute to both periods leading to different ensembles being used to compute these estimates. Despite the different ensembles (seven studies for 2002-2006 and ten studies for 2008-2012), the estimate range for both periods are similar. Keeping only the five surface-based inversions covering both periods leads to 542 [530-554] Tg $CH_4$ yr$^{-1}$ on average over 2002-2006 and 563 [546-573] Tg $CH_4$ yr$^{-1}$ over 2008-2012, showing remarkably consistent values with the

ensemble of the top-down studies and also not showing significant impact in the emission differences between the two time periods (see Table S3).





The emission changes between the period 2002-2006 and the period 2008-2012 have been calculated for inversions covering at least three years over both 5-year period (5 inversions) at global, hemispheric, and regional scales (Fig. 4). The regions are the same as in Saunois et al. (2016). The region denoted as " 90°S-30°N" is referred as the tropics despite the southern mid-latitudes (mainly from Oceania and temperate South America) included in this region. However, since the extra tropical

Southern Hemisphere contributes less than 8% to the emissions from the "90°S-30°N" region, the region represents primarily the tropics.

The global emission increase of +22 [16-32] Tg $CH_4$ $yr^{-1}$ is mostly tropical (+18 [13-24] Tg $CH_4$ $yr^{-1}$, or ~80% of the global increase). The northern mid-latitudes only contribute an increase of +4 [0-9] Tg $CH_4$ $yr^{-1}$, while the high-latitudes (above 60°N) contribution is not significant. Yet most of inversions rely on surface observations, which poorly represent the tropical

continents. As a result, this tropical signal may partly be an artefact of inversions attributing emission changes to unconstrained regions. Also the absence of a significant contribution from the Arctic region means that Arctic changes are below the detection limit of inversions. Indeed, the northern high latitudes emitted about 20 [14-24] Tg $CH_4$ $yr^{-1}$ of methane over 2002-2006 and 22 [15-31] Tg $CH_4$ $yr^{-1}$ over 2008-2012 (Table 3); but keeping inversions covering at least three years over each 5-year period leads to a null emission change in boreal regions.

The geographical partition of the increase in emissions between 2000-2006 and 2008-2012 inferred here is in agreement with Bergamaschi et al. (2013) who found that 50-85 % of the 16-20 Tg $CH_4$ emission increase between 2007-2010 compared to 2003-2005 came from the tropics and the rest from the northern hemisphere mid-latitudes. Houweling et al. (2014) inferred an increase of 27-35 Tg $CH_4$ $yr^{-1}$ between the 2-years periods before and after July 2006, respectively. The ensemble of inversions gathered in this study infers a consistent increase of 30 [20-41] Tg $CH_4$ $yr^{-1}$ between the same two periods. The

derived increase is highly sensitive to the choice of the month beginning and ending the period. The study of Patra et al. (2016) based on six inversions found an increase of 19-36 Tg $CH_4$ $yr^{-1}$ in global methane emissions between 2002-2006 and 2008-2012, which is consistent with our results.

### 3.2.2 Regional changes inferred by top-down inversions

At the regional scale, top-down approaches infer different emission changes both in amplitude and in sign. These

discrepancies are due to transport errors in the models and to differences in inverse setups, and can lead to several tens of per cent of differences in the regional estimates of methane emissions (e.g., Locatelli et al. (2013)). Indeed, the recent study of Cressot et al. (2016) showed that, while global and hemispheric emission changes could be detected with confidence by the top-down approaches using satellite observations, their regional attribution is less certain. Thus it is particularly critical for regional emissions to rely on several inversions, as done in this study, before drawing any robust conclusion. In most of the

top-down results (Fig. 4), the tropical contribution to the global emission increase originates mainly from tropical South America (+9 [6-13] Tg $CH_4$ $yr^{-1}$) and, South and East Asia (+5 [-6-10] Tg $CH_4$ $yr^{-1}$). Central North America (+2 [0-5] Tg $CH_4$ $yr^{-1}$) and Northern Africa (+2 [0-5] Tg $CH_4$ $yr^{-1}$) contribute less to the tropical emission increase. The sign of the contribution from South and East Asia is positive in most studies (e.g., Houweling et al. (2014)), although some studies infer





decreasing emission in this region. The disagreement between inversions could result from the lack of measurement stations to constrained the fluxes in Asia (some have appeared inland India and China but only in the last years, Lin et al. (2017)), and also from the rapid up-lift of the compounds emitted at the surface to the free troposphere by convection in this region, leading to surface observations missing information on local fluxes (e.g., Lin et al. (2015)).

In the northern mid-latitudes a positive contribution is inferred for China (+4 [1-11] Tg $CH_4$ $yr^{-1}$) and Central Eurasia and Japan (+1 [-1-6] Tg $CH_4$ $yr^{-1}$). Also, temperate North America does not contribute significantly to the emission changes, as none of the inversions detect, at least prior to 2013, an increase in methane emissions due to increasing shale gas exploitation in the U.S. (Bruhwiler et al., accepted).

The inversions do not agree on the sign of the emission change over the high northern latitudes, especially over boreal North
America; however, they show a consistent small emission decrease in Russia. This lack of agreement between inversions over the boreal regions highlights the weak sensitivity of inversions in these regions where no or little methane emission changes occurred over the last decade. Changes in wetland emissions associated with sea ice retreat in the Arctic are probably only a few Tg between the 1980s and the 2000s (Parmentier et al., 2015). Also decreasing methane emissions in sub-Arctic areas that were drying and cooling over 2003-2011 have offset increasing methane emissions in a wetting Arctic
and warming summer (Watts et al., 2014). Despite a small increase in late autumn/early winter in methane emission from Arctic tundra, no significant long-term trends in methane emission have been observed yet (Sweeney et al., 2016). However, unintentional double counting of emissions from different water systems (wetlands, rivers, lakes) may lead to Artic emission growth when little or none exists (Thornton et al., 2016).

### 3.2.3 Emission changes in bottom-up studies.

The top-down approaches use bottom-up estimates as *a priori* values. For anthropogenic emissions, most of them use the EDGARv4.2FT2010 inventory and GFED3 emission estimates for biomass burning. Their source of priori information differs more for the contribution from natural wetlands, geological emissions, and termite sources (Table S1). Here we gathered an ensemble of bottom-up estimates for the changes in methane emissions between 2000-2006 and 2008-2012 combining anthropogenic inventories (EDGARv4.2FT2010, USEPA and GAINS), five biomass burning emission estimates
(GFED3, GFED4s, FINN, GFAS and FAOSTAT) and wetland emissions from eleven land surface models (see Sect. 2 for the details and in Saunois et al. (2016) and Poulter et al. (in review)). As previously stated, other natural methane emissions (termites, geological, inland waters) are assumed not to contribute significantly to the change between 2000-2006 and 2008-2012, because no quantitative indications are available on such changes and because at least some of these sources are less climate-sensitive than wetlands.

The bottom-up estimate of the global emission change between the periods 2000-2006 and 2008-2012 (+21 [5-41] Tg $CH_4$ $yr^{-1}$, Fig. 4) is comparable but possesses with a larger spread than top-down estimates (+22 [16-32] Tg $CH_4$ $yr^{-1}$). Also, the hemispheric breakdown of the change reveals discrepancies between top-down and bottom-up estimates. The bottom-up approaches suggest much higher increase of emissions in the mid latitudes (+17 [6-30] Tg $CH_4$ $yr^{-1}$) than inversions and a





smaller increase in the tropics (+6 [-4-13] Tg $CH_4$ $yr^{-1}$). The main regions where bottom-up and top-down estimates of emission changes differ are tropical South America, South and East Asia, China, USA, and central Eurasia and Japan.

While top-down studies indicate a dominant increase between 2000-2006 and 2008-2012 in tropical South America (+9 [6-13] Tg $CH_4$ $yr^{-1}$), the bottom-up estimates (based on an ensemble of 11 land surface models and anthropogenic inventories),

in contrast, indicate a small decrease (-2 [-6-2] Tg $CH_4$ $yr^{-1}$) over the same period (Fig. 4). The decrease in tropical South American emissions found in the bottom-up studies results from decreasing emissions from wetlands (about -2.5 Tg $CH_4$ $yr^{-1}$, mostly due to a reduction in tropical wetland extent) and biomass burning (about -0.7 Tg $CH_4$ $yr^{-1}$), partly compensated by a small increase in anthropogenic emissions (about 1 Tg $CH_4$ $yr^{-1}$, mainly from agriculture and waste). Most of the top-down studies infer a decrease in biomass burning emissions over this region, exceeding the decrease in a priori emissions from

GFED3. Thus the main discrepancy between top-down and bottom-up is due to microbial emissions from natural wetlands (about 4 Tg $CH_4$ $yr^{-1}$ on average), agriculture and waste (about 2 Tg $CH_4$ $yr^{-1}$ on average) over tropical South America.

The emission increase in South and East Asia for the bottom-up estimates (2 Tg $CH_4$ $yr^{-1}$) results from a 4 Tg $CH_4$ $yr^{-1}$ increase (from agriculture and waste for half of it, fossil fuel for one third and wetland for the remainder) offset by a decrease in biomass burning emissions (-2 [-4-0] Tg $CH_4$ $yr^{-1}$). The inversions suggest a higher increase in South and East

Asia compared to this 2 Tg $CH_4$ $yr^{-1}$, mainly due to higher increases in wetland and agriculture and waste sources; the biomass burning decrease and the fossil fuel increase being similar in the inversions compared to the inventories.

In tropical South America and South and East Asia, wetlands and agriculture and waste emissions partly occur over same areas, making the partitioning difficult for the top-down approaches. Also, these two regions lack of surface measurement sites, so that the inverse systems are less constrained by the observations. However, the SCIAMACHY-based inversion from

Houweling et al. (2014) also infers increasing methane emissions over tropical South America between 2002-2006 and 2008-2012. Further studies based on satellite data or additional regional surface observations (e.g., Basso et al. (2016); Xin et al. (2015)) would be needed to better assess methane emissions (and their changes) in these under-sampled regions.

For China, bottom-up approaches suggest a +10 [2-20] Tg $CH_4$ $yr^{-1}$ emission increase between 2002-2006 and 2008-2012, which is much larger than the top-down estimates. The magnitude of the Chinese emission increase varies among emission

inventories and appears essentially to be driven by an increase in anthropogenic emissions (fossil fuel and agriculture and waste emissions). Anthropogenic emission inventories indicate that Chinese emissions increased at a rate of 0.6 Tg $CH_4$ $yr^{-2}$ in USEPA, 3.1 Tg $yr^{-2}$ in EDGARv4.2 and 1.5 Tg $yr^{-2}$ in GAINS between 2000 and 2012. The increase rate in EDGARv4.2 is too strong compared to a recent bottom-up study that suggests a 1.3 Tg $CH_4$ $yr^{-2}$ increase in Chinese methane emissions over 2000-2010 (Peng et al., 2016). The revised EDGAR inventory v4.3.2 (not released yet) with region-specific

emission factors for coal mining in China gives a mean trend in coal emissions of 1.0 Tg $CH_4$ $yr^{-2}$ over 2000-2010, half the value from the previous version EDGARv4.2FT2010 (Fig. S14). These new estimates are more in line with USEPA inventory and with the top-down approaches (range of 0.3 to 2.0 Tg $CH_4$ $yr^{-2}$ for the total sources in China over 2000-2012), in agreement with Bergamaschi et al. (2013) who inferred an increase rate of 1.1 Tg $CH_4$ $yr^{-2}$ over 2000-2010.




Finally, while bottom-up approaches show a small increase in U.S. emissions (+2 [-1-4] Tg CH$_4$ yr$^{-1}$), top-down studies do not show any significant emission change, similarly for central Eurasia and Japan.

### 3.2.4 Emission changes by source types

In Sect. 3.1, we suggest that a concurrent increase in both natural and anthropogenic emissions over 2006-2008 contribute to
the total emission increase between 2002-2006 and 2008-2012. The attribution of this change to different source types remains uncertain in inversions, as methane observations alone do not provide sufficient information to fully separate individual sources (see Introduction). Yet, as in Saunois et al. (2016), we present here a sectorial view of methane emissions for five general source categories, limited at the global scale (Fig. 5), as regional attribution of emission increase is considered too uncertain (Saunois et al., 2016; Tian et al., 2016).

The top-down studies show a dominant positive contribution from microbial sources (agriculture and waste (+10 [7-12] Tg CH$_4$ yr$^{-1}$ and natural wetlands (+6 [-4-16] Tg CH$_4$ yr$^{-1}$) as compared to fossil fuel related emissions (+7 [-2-16] Tg CH$_4$ yr$^{-1}$),. Biomass burning emissions decreased (-3 [-7-0] Tg CH$_4$ yr$^{-1}$). Other natural sources show a lower but significant increase (+2 [-2-7] Tg CH$_4$ yr$^{-1}$). These values are estimated based on the five longest inversions. Taking into account shorter inversions leads to different minimum and maximum values, but the mean values are quite robust (Table S4).

Wetland emission changes estimated by 11 land surface models from Poulter et al. (in review) are near zero but the stability of this source is statistically consistent with the top-down value considering the large uncertainties of both top-down inversions and bottom-up models (Sect. 3.1 and Sect. 4 Discussion). It is worth noting that, for wetland prior estimates, top-down studies generally rely on climatology from bottom-up approaches (e.g., Matthews and Fung (1987); Kaplan (2002)) and therefore the inferred trend are more independent from bottom-up models than anthropogenic estimates, which generally
relies on inter-annually prescribed prior emissions.

The bottom-up estimated decrease in biomass burning emissions of (-2 [-5-0] Tg CH$_4$ yr$^{-1}$) is consistent with top-down estimates albeit smaller. The change in agriculture and waste emissions between 2002-2006 and 2008-2012 in the bottom up inventories are in agreement with the top-down values (+10 [7-13] Tg CH$_4$ yr$^{-1}$), with about two-third of this being increase from agriculture activities (mainly enteric fermentation and manure management, while rice emissions were fairly constant
between these two time periods) and one-third from waste (Table S5). The spread between inventories in the increase of methane emissions from the waste sector is much lower than from agriculture activities (enteric fermentation and manure management, and rice cultivation) (see Table S5). Considering livestock (enteric fermentation and manure) emissions estimated by FAOSTAT, about half of the global increase between 2002-2006 and 2008-2012 originates from Asia (India, China and, South and East Asia) and one-third from Africa.

The changes in fossil fuel related emissions in bottom-up inventories between 2002-2006 and 2008-2012 (+17 [11-25] Tg CH$_4$ yr$^{-1}$) are more than twice the estimate from the top-down approaches (+7 [-2-16] Tg CH$_4$ yr$^{-1}$). Among the inventories, EDGARv4.2 stands in the higher range, with fossil fuel related emissions increasing twice as fast as in USEPA and GAINS. The main contributors to this discrepancy are the emissions from coal mining, which increase at three times as fast as in





EGDARv4.2 than in the two other inventories at the global scale. About half of the global increase in fossil fuel emissions originates from China in the EDGARv4.2 inventory. Thus, most of the difference between top-down and bottom-up originates from coal exploitation estimates in China, which is likely overestimated in EDGARv4.2 as aforementioned (Bergamaschi et al., 2013; Peng et al., 2016; Dalsoren et al., 2016; Patra et al., 2016; Saunois et al., 2016). The release of

EDGARv4.3.2 will, at least partly, close the gap between top-down and bottom-up studies. Indeed, in EDGARv4.3.2 coal emissions in China increase by 4.3 Tg $CH_4$ $yr^{-1}$ between 2002-2006 and 2008-2010 instead of 9.7 Tg $CH_4$ $yr^{-1}$ in EDGARv4.2FT2010, due to the revision of coal emission factors in China. As a result, the next release of EDGARv4.3.2 should narrow the range and decrease the mean contribution of fossil fuels to emission changes estimated by the bottom-up studies.

**4 Discussion**

The top-down results gathered in this synthesis suggest that the emission increase in methane emissions between 2002-2006 and 2008-2012 is mostly tropical, with a small contribution from the mid-latitudes, and is dominated by an increase in microbial sources, more from agriculture and waste (+10 [7-12] Tg $CH_4$ $yr^{-1}$) than wetlands, the latter being uncertain (+6 [-4-16] Tg $CH_4$ $yr^{-1}$). The contribution from fossil fuels to this emission increase is uncertain but smaller on average (+7 [-2-

16] Tg $CH_4$ $yr^{-1}$). These increases in methane emissions are partly counterbalanced by a decrease in biomass burning emissions (-3 [-7-0] Tg $CH_4$ $yr^{-1}$). These results are in agreement with the top-down studies of Bergamaschi et al. (2013) and Houweling et al. (2014), though there are some discrepancies between inversions in the regional attribution of the changes in methane emissions. The sectorial partitioning from inversions is in agreement (within the uncertainty) with bottom-up inventories (noting that inversions are not independent from inventories), though the top-down ensemble significantly

decreases the methane emission change from fossil fuel production and use compared to the bottom-up inventories, although the estimate of the latter should decrease with the upcoming revised version of the EDGAR inventory (see Sect. 3.2.4).

**Wetland contribution.** The increasing emissions from natural wetlands inferred from the top-down approaches are not consistent with the average of the land surface models from Poulter et al. (in review). Bloom et al. (2010) found that wetland methane emissions increased by 7% over 2003-2007 mainly due to warming in the mid-latitudes and Arctic regions and that

tropical wetland emissions remained constant over this period. Increases of 2 [-1-5] Tg $CH_4$ $yr^{-1}$ and of 1 [0-2] Tg $CH_4$ $yr^{-1}$ between 2002-2006 and 2008-2012 are inferred from the eleven land surface models over the northern mid-latitudes and boreal regions, respectively (Table S7, linked to temperature increase). Decreasing wetland emissions in the tropics (mostly due to reduced wetland extent) in the land surface models (-3 [-8-0] Tg $CH_4$ $yr^{-1}$) offset the mid-latitude and boreal increases, resulting in stable emissions between 2002-2006 and 2008 at the global scale. These different conclusions between

inversions and wetland models highlight the difficulties in estimating wetland methane emissions (and their changes). Also the spread of land surface models driven with the same flooded area extent shows that the models are highly sensitive to the wetland extent, temperature, precipitation, and atmospheric $CO_2$ feedbacks (Poulter et al., in review). The JULES land model



used by McNorton et al. (2016b) is one of the three models inferring slightly higher emissions in 2008-2012 than 2002-2006 from the ensemble used in our study (Table S6). Yet, they found larger increases in northern mid-latitude wetland emissions and near zero change in tropical wetland emissions, contrary to the atmospheric inversions. The exponential temperature dependency of methanogenesis through microbial production has been recently revised upwards (Yvon-Durocher et al.,

2014). Accounting for this revision, smaller temperature increases are needed to explain large methane emission changes in warm climate (such as in the tropics) (Marotta et al., 2014). However, no significant trend in tropical surface temperature is inferred over 2000-2012 (Poulter et al., in review) that could explain an increase in tropical wetland emissions per Meter Square. Methane emissions are also sensitive to the extent of the flooded area and for non-flooded wetlands, and to the depth of the water table (Bridgham et al., 2013). The recurrent La Niña conditions from 2007 (compared to more El Niño

conditions in the beginning of the 2000s) may have triggered wetter conditions propitious to higher methane emissions in the tropics (Nisbet et al., 2016). Indeed, both the flooded data set used in Poulter et al. (in review) and the one used in Mc Norton et al. (2016b) based on an improved version of the TOPography-based hydrological MODEL (Marthews et al., 2015), show decreasing wetland extents from the 2000s to the 2010s. However resulting decreasing methane emissions are not in agreement with top-down studies even when constrained by satellite data. Thus, as has been concluded in most land

model CH$_4$ inter-comparisons and analyses, more efforts are needed to better assess the wetland extent and its variations (e.g., Bohn et al. (2015); Melton et al. (2013); Xu et al. (2016)). Even though top-down approaches may incorrectly attribute the emissions increase between 2002-2006 and 2008-2012 to tropical regions (and hence partly to wetland emitting areas) due to a lack of observational constraints, it is not possible, with the evidence provided in this study, to rule out a potential positive contribution of wetland emissions in the increase of global methane emissions at the global scale.

**Isotopic constraints**. The recent variation in atmospheric methane mole fractions has been widely discussed in the literature in relation with concurrent methane isotopes. Schaefer et al. (2016) tested several scenarios of perturbed methane emissions to fit both atmospheric methane and δ$^{13}$C-CH$_4$. For the post 2006 period (2007-2014), they found that an average emission increase of 19.7 Tg CH$_4$ yr$^{-1}$ with an associated isotopic signature of about -59 ‰ (-61 ‰ to -56 ‰) is needed to match both CH$_4$ and δ$^{13}$C-CH$_4$ observed trends. After assigning an isotopic signature (δ$_i$) of each source contribution to the change (ΔE$_i$),

it is possible to estimate the average isotopic signature of the emission change (δ$_{ave}$) as the weighted mean of the isotopic signature of all the sources contributing to the change, following Equation 1:

$$\delta_{ave} = \frac{1}{\sum_i \Delta E_i} \sum_i \delta_i \Delta E_i \qquad\qquad (1)$$

However, assigning an isotopic signature to a specific source remains a challenge due to sparse sampling of the different sources and wide variability of the isotopic signature of each given source: for example methane emissions from coal mining

have a range of -70 ‰ to -30 ‰ in δ$^{13}$C-CH$_4$ (Zazzeri et al., 2016; Schwietzke et al., 2016). The difficulty increases when trying to assign an isotopic signature to a broader category of methane sources at the global scale. Schaefer et al. (2016) suggest the following global mean isotopic signatures: -60‰ for microbial sources (wetland, agriculture and waste), -37‰ for thermogenic (fossil fuel sources) and -22‰ for pyrogenic (biomass burning emissions); while a recent study suggests



different globally averaged isotopic signatures, with a lighter fossil fuel signature: -44‰ for fossil fuels, -62‰ for microbial, and -22‰ for biomass burning emissions (Schwietzke et al., 2016). Also there is the question on the isotopic signature to be attributed to "other natural" sources that include geological emissions (~-49‰, Etiope (2015)), termites (~-57‰, Houweling et al. (2000)), or oceanic sources (~-40‰, Houweling et al. (2000)). Applying either set of isotopic signature to the bottom-

up estimates of methane emission changes leads to (expected) unrealistically heavy $\delta^{13}CH_4$ signatures due to large changes in fossil fuel emissions (Fig. 6). Most of the individual inversions do not agree with the atmospheric isotopic change between 2002-2006 and 2008-2012 (Fig. 6), due to their large increases in fossil fuel or wetland emissions and/or large decrease in biomass burning emissions (Table S4). Most of the inverse systems solve only for total net methane emissions making the sectorial partition uncertain and dependent on the prior partitioning. However, applying Schaefer et al. (2016) isotopic

source signatures to the mean emission changes derived from the ensemble of inversions in Eq. 1 leads to an average isotopic signature of the emission change well in agreement with the range of Schaefer et al. (2016), whatever the choice made for the "other natural" sources or the number of inversions selected (Fig. 6). Applying Schwietzke et al. (2016) isotopic source signatures leads to lighter average isotopic signature of the emission change, in the higher range (in absolute value) of Schaefer et al. (2016). In short, the isotopic signature of the emissions change between 20002-2006 and 2008-2012

derived from the ensemble mean of inversions seem consistent with $^{13}C$ atmospheric signals. Yet the uncertainties of these mean emission changes remain very large as shown by the range inferred by inversions. Also, the deviations of most of the individual inversions from the ensemble mean highlight the sensitivity of the atmospheric isotopic signal to the changes in methane sources. To conclude, isotopic studies such as Schaefer et al. (2016) can help eliminate combinations of sources that are unrealistic, but cannot point towards a unique solution. This problem has more unknowns than constraints and other

pieces of information need to be added to further solve it (such as $^{14}C$, deuterium, or co-emitted species).

**Ethane constraint.** Co-emitted species with methane, such as ethane from fugitive gas leaks, can also help in assessing contributions from oil and gas sources. Indeed, Haussmann et al. (2016) used ethane to methane emission ratios to estimate the contribution from oil and gas emissions to the recent methane increase. For 2007-2014, their emission optimization suggests that total methane emissions increased by 24-45 Tg $CH_4$ $yr^{-1}$, which is larger than in our study (Sect. 3.2.1), but the

time period covered only partially overlaps with our study and they use a different method. Assuming a linear trend over 2007-2014 leads to an increase of 18-34 Tg $CH_4$ $yr^{-1}$ over 2007-2012. Their reference scenario assumes that a mixture of oil and gas sources contributed at least 39% of the increase of total emissions, corresponding to an increase in oil and gas methane emissions of 7-13 Tg $CH_4$ $yr^{-1}$ over 2007-2012. Adding up the increase in methane emissions from coal mining (USEPA suggests a 4 Tg $CH_4$ $yr^{-1}$ increase between 2002-2006 and 2008-2012, Table S5) would lead to an increase in fossil

fuel emission in the upper range of the top-down estimates presented here (7 [-2-16] Tg $CH_4$ $yr^{-1}$). Helmig et al. (2016), using a ethane to methane emission ratio of 10% and assuming it constant, calculated an increase of 4.4 Tg $CH_4$ $yr^{-1}$ each year during 2009-2014, which leads to a cumulative increase inconsistent in regards with both the global atmospheric isotopic signal and the observed leak rates in productive regions. Ethane to methane emission ratios are uncertain (ranging 7.1 to 16.2% in Haussmann et al. (2016) reference scenario and 16.2 to 32.4 % in their pure oil scenario) and could



experienced variations (e.g., Wunch et al. (2016)) that are not taken into account due to lack of information. Indeed, ethane to methane emission ratios also largely depends on the shale formation and considering a too low ethane to methane emission ratio could lead to erroneously too large methane emissions from shale gas (Kort et al., 2016). Besides, the recent bottom-up study of Höglund-Isaksson (2017) shows relatively stable methane emissions from oil and gas after 2007, due to

increases in recovery of associated petroleum gas (particularly in Russia and Africa) that balances an increase in methane emissions from unconventional gas production in North America.

Overall, the mean emission changes resulting from the top-down approach ensemble agree well with the isotopic atmospheric observations but further studies (inversions and field measurements) would be needed to consolidate the (so far) weak agreement with the ethane-based global studies. Better constraints on the relative contributions of microbial emissions

and thermogenic emissions derived from the top-down approaches using both isotopic observations and additional measurements such as ethane (with more robust emission ratios to methane) or other hydrocarbons (Miller et al., 2012) would help improve the ability to separate sources using top-down inversions.

**Methane sink by OH.** As stated in Sect.2, this paper focuses on methane emission changes. The methane sinks, especially OH oxidation, can also play a role in the methane budget changes. However the results from the inversions presented here,

for most of them, assume constant OH concentrations over the period 2000-2012 (though including seasonal variations, Table S2). Before 2007, increasing OH concentrations could have contributed to the stable the atmospheric methane burden in this period (Dalsøren et al., 2016), without (or with less of) a need for constant global emissions. Including OH variability in their tests, Schaefer et al. (2016) found that $CH_4$ variations can be explained only up to 2008 by changes in OH only and that an isotopic signature of the total additional source of -65‰ is necessary to explain the $\delta^{13}C$-$CH_4$ observations (see their

supplementary materials). However a -65‰ isotopic signature of additional emissions would require even less changes from fossil fuel emissions or more changes from microbial. After 2007, McNorton et al. (2016a), based on methyl chloroform measurements, found that global OH concentrations decreased after 2007 (up to -6% between 2005 and 2010, their Fig 1.d). Consistently, Dalsøren et al. (2016) suggested that the recent methane increase is due first to high emissions in 2007-2008 followed by a stabilization in methane loss due to meteorological variability (warm year 2010), both leading to an increase in

methane atmospheric burden. In this context, decreasing OH concentrations alone does not seem sufficient to explain all of the recent methane increase. However, decreasing OH concentrations since 2008 would require smaller emission changes to explain the observed atmospheric methane increase, also possibly implying a different partitioning of emission types to match the atmospheric $\delta^{13}C$ evolution. In a scenario where OH decreases, however, the disagreement between the top-down and bottom-up estimated emission change between 2002-2006 and 2008-2012 would increase.

**5 Conclusions**

Following the decadal methane budget published by Saunois et al. (2016) for the time period 2000-2012, variations of methane sources over the same period are synthesized from an ensemble of top-down and bottom-up approaches gathered





under the umbrella of The Global Carbon Project – Global Methane Budget initiative. The mean top-down model ensemble suggests that annual global methane emissions have increased between 2000 and 2012 by 15-33 Tg $CH_4$ $yr^{-1}$ with a main contribution from the tropics, with additional emissions from the mid-latitudes, but showing no signal from high latitudes. We suggest that global methane emissions have experienced a shift between 2006 and 2008 resulting from an increase in

both natural and anthropogenic emissions. Based on the top-down ensemble mean, during 2000-2006, increasing anthropogenic emissions were compensated by decreasing natural emissions and, during 2008-2012, both anthropogenic and natural emissions were rather stable.

To further investigate the apparent source shift, we have analyzed the emission changes between 2002-2006 and 2008-2012. The top-down ensemble mean shows that annual global methane emissions increased by 20 [13-32] Tg $CH_4$ $yr^{-1}$ between

these two time periods with the tropics contributing about 80% to this change, and the remainder coming from the mid-latitudes. The regional contributions are more uncertain, especially in the tropics where tropical South America and, South and East Asia are the main contributors, although contrasting contributions from South East Asia among inversions are inferred. Such regional uncertainties are due to a lack of measurements from surface stations in key tropical regions, forcing inversion systems to estimate emissions in regions without observational constraints. A consistent result among the top-down

inverse models is that their inferred global emission increases are much lower than those estimated from the bottom-up approaches. This is particularly due to an overestimation of the increase in the anthropogenic emissions from China.

As methane atmospheric observations alone cannot be used to fully distinguish between methane emission processes, sectorial estimates have been reported for only five broad categories. The ensemble of top-down studies gathered here suggests a dominant contribution to the global emission increase from microbial sources (+16 Tg $CH_4$ $yr^{-1}$ with +10 [7-12]

Tg $CH_4$ $yr^{-1}$ from agriculture and waste, and +6 [-4-16] Tg $CH_4$ $yr^{-1}$ from wetlands), and an uncertain but smaller contribution of +7 [-2-16] Tg $CH_4$ $yr^{-1}$ from fossil fuel related emissions from 2000-2006 to 2008-2012. In the top-down ensemble, biomass burning emissions decreased by -3 [-7-0] Tg $CH_4$ $yr^{-1}$. Interestingly, the magnitudes of these mean changes for individual source sectors based on ensemble mean results from top-down approaches are consistent with isotopic observations (Schaefer et al., 2016), while the individual inversions are generally not. Yet the uncertainties of these mean

emission changes are very large as shown by the range inferred by inversions.

The interpretation of changes in atmospheric methane in this study is limited mostly to changes in terms of changes in methane emissions. The results from the inversions presented here mostly assume constant OH concentrations over the period 2000-2012 (though including seasonal variations, Table S2). As a result, changes in methane loss through OH oxidation in the atmosphere and soil uptake of methane, are not addressed here, and their contribution needs to be further

investigated to better understand the observed growth rate changes during the analysed period. Indeed, the inferred shift in emissions during 2006-2008 could likely be much smoother if OH concentrations decreased during these three years after a period of increase, as suggested in recent studies (e.g., Dalsoren et al. (2016)). Estimating and optimizing OH oxidation in top-down approaches is challenging due to uncertainties in the 4D fields of OH concentrations used by the models. Although beneficial for the recovery of the stratospheric ozone, methyl-chloroform, which is used as a proxy to derive OH variations,



is disappearing from the atmosphere, and as a result, is becoming much less useful for inferring OH concentration changes. This also implies that we need new proxies to infer and constrain global OH concentrations. Chemistry climate models may be useful to provide OH 4D fields and to estimate its impact on lifetime, though large discrepancies exist, especially at the regional scale (Naik et al., 2013).

The global methane budget is far from being understood to the same level of detail as the $CO_2$ budget currently is. Indeed, the recent acceleration of the methane atmospheric growth rate in 2014 and 2015 (Ed Dlugokencky, NOAA/ESRL (www.esrl.noaa.gov/gmd/ccgg/trends_ch4/) adds more challenges to our understanding of the methane global budget. The next Global Methane Budget will aim to include data from these recent years and make use of additional surface observations from different tracers, and satellite data to better constrain the time evolution of atmospheric methane burden.

**Acknowledgements**

This collaborative international effort is part of the Global Carbon Project activity to establish and track greenhouse gas budgets and their trends. M. Saunois and P. Bousquet acknowledge the Global Carbon Project for the scientific advice and the computing support of LSCE-CEA and of the national computing center TGCC.

B. Poulter has been funded by the EU FP7 GEOCARBON project. J. G. Canadell thanks the support from the National Environmental Science Program – Earth Systems and Climate Change Hub. D.R. Black and I.J. Simpson (UCI) acknowledge funding support from NASA (NNX07AK10G). F. Joos, R. Spahni, and R. Schroeder acknowledge support by the Swiss National Science Foundation. C. Peng acknowledges the support by National Science and Engineering Research Council of Canada (NSERC) discovery grant and China's QianRen Program. G. P. Peters acknowledges the support of the

Research Council of Norway project 209701. D. Bastviken acknowledge support from the Swedish Research Council VR and ERC (grant no. 725546). Patrick Crill acknowledge support from the Swedish Research Council VR. F. N. Tubiello acknowledges the support of FAO Regular Programme Funding under O6 and SO2 for the developed and maintenance of the FAOSTAT Emissions database. The FAOSTAT database is supported by regular programme funding from all FAO member countries. P.K. Patra is partly supported by the Environment Research and Technology Development Fund (A2-1502) of the

Ministry of the Environment, Japan. W.J. Riley and X. Xu were supported by the Director, Office of Science, Office of Biological and Environmental Research of the US Department of Energy under Contract DE-AC02-05CH11231 as part of the RGCM BGC-Climate Feedbacks SFA. P. Bergamaschi and M. Alexe acknowledge the support by the European Commission Seventh Framework Programme (FP7/2007–2013) project MACCII under grant agreement 283576, by the European Commission Horizon2020 Programme project MACC-III under grant agreement 633080, and by the ESA Climate

Change Initiative Greenhouse Gases Phase 2 project. H. Tian and B. Zhang acknowledge support by NASA Carbon Monitoring Program (NNX12AP84G, NNX14AO73G). H.-S. Kim and S. Maksyutov acknowledge use of the GOSAT Research Computation Facility. N. Gedney and A. Wiltshire acknowledge support by the Joint DECC/Defra Met Office Hadley Centre Climate Programme (GA01101). D. J. Beerling acknowledges support from an ERC Advanced grant (CDREG, 322998) and NERC (NE/J00748X/1).





The CSIRO and the Australian Government Bureau of Meteorology are thanked for their ongoing long-term support of the Cape Grim station and the Cape Grim science programme. The CSIRO flask network is supported by CSIRO Australia, Australian Bureau of Meteorology, Australian Institute of Marine Science, Australian Antarctic Division, NOAA USA, and the Meteorological Service of Canada. The operation of the AGAGE instruments at Mace Head, Trinidad Head, Cape Matatula, Ragged Point, and Cape Grim is supported by the National Aeronautic and Space Administration (NASA) (grants NAG5-12669, NNX07AE89G, and NNX11AF17G to MIT and grants NNX07AE87G, NNX07AF09G, NNX11AF15G, and NNX11AF16G to SIO), the Department of Energy and Climate Change (DECC, UK) contract GA01081 to the University of Bristol, and the Commonwealth Scientific and Industrial Research Organization (CSIRO Australia), and Bureau of Meteorology (Australia).

M. Saunois and P. Bousquet acknowledge Lyla Taylor (University of Sheffield, UK), Chris Jones (Met Office, UK) and Charlie Koven (Lawrence Berkeley National Laboratory, USA) for their participation to land surface modelling of wetland emissions. Theodore J. Bohn (ASU, USA), Jens Greinhert (GEOMAR, the Netherlands), Charles Miller (JPL, USA), and Tonatiuh Guillermo Nunez Ramirez (MPI Jena, Germany) are thanked for their useful comments and suggestions on the manuscript. M. Saunois and P. Bousquet acknowledge Martin Herold (WU, the Netherlands), Mario Herrero (CSIRO, Australia), Paul Palmer (University of Edinburgh, UK), Matthew Rigby (University of Bristol, UK), Taku Umezawa (NIES, Japan), Ray Wang (GIT, USA), Jim White (INSTAAR, USA), Tatsuya Yokota (NIES, Japan), Ayyoob Sharifi and Yoshiki Yamagata (NIES/GCP, Japan) and Lingxi Zhou (CMA, China) for their interest and discussions on the Global Carbon project methane. M. Saunois and P. Bousquet acknowledge the initial contribution to the Global Methane Budget 2016 release and/or possibly future contribution to the next Global Methane Budget of Victor Brovkin (MPI Hamburg, Germany), Charles Curry (University of Victoria, Canada), Kyle C. McDonald (City University of New-York, USA), Julia Marshall (MPI Jena, Germany), Christine Wiedinmyer (NCAR, USA), Michiel van Weele (KNMI, Netherlands), Guido R. van der Werf (Amsterdam, Netherlands) and Paul Steele (retired from CSIRO, Australia).

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





**Table 1: List of the top-down estimates included in this paper.**

| Model | Institution | Observation used | Time period | Flux solved | Number of inversions | References |
|---|---|---|---|---|---|---|
| Carbon Tracker-CH$_4$ | NOAA | Surface stations | 2000-2009 | 10 terrestrial sources and oceanic source | 1 | Bruhwiler et al. (2014) |
| LMDZ-MIOP | LSCE/CEA | Surface stations | 1990-2013 | Wetlands, biomass burning and other natural, anthropogenic sources | 10 | Pison et al. (2013) |
| LMDZ-PYVAR | LSCE/CEA | Surface stations | 2006-2012 | Net source | 6 | Locatelli et al. (2015) |
| LMDZ-PYVAR | LSCE/CEA | GOSAT satellite | 2010-2013 | | 3 | |
| TM5 | SRON | Surface stations | 2003-2010 | | 1 | Houweling et al. (2014) |
| TM5 | SRON | GOSAT satellite | 2009-2012 | Net source | 2 | |
| TM5 | SRON | SCIAMACHY satellite | 2003-2010 | | 1 | |
| TM5 | EC-JRC | Surface stations | 2000-2012 | Wetlands, rice, biomass burning and all remaining sources | 1 | Bergamaschi et al. (2013), Alexe et al. (2015) |
| TM5 | EC-JRC | GOSAT satellite | 2010-2012 | | 1 | |
| GELCA | NIES | Surface stations | 2000-2012 | Natural (wetland, rice, termite), anthropogenic (excluding rice), biomass burning, soil sink | 1 | Ishizawa et al. 2016;Zhuravlev et al. (2013) |
| ACTM | JAMSTEC | Surface stations | 2002-2012 | Net source | 1 | Patra et al. (2016) |
| NIES-TM | NIES | Surface stations | 2010-2012 | Biomass burning, anthropogenic emissions (excluding rice paddies) and all natural sources (including rice paddies) | 1 | Kim et al. (2011), Saito et al. (2016) |
| NIES-TM | NIES | GOSAT satellite | 2010-2012 | | 1 | |



**Table 2: List of the bottom-up studies included in this paper.**

| B-U models and inventories | Contribution | Time period (resolution) | Gridded | References |
|---|---|---|---|---|
| EDGAR4.2 FT2010 | Fossil fuels, Agriculture and waste, biofuel | 2000-2010 (yearly) | X | EDGARv4.2FT2010 (2013), Olivier et al. (2012) |
| EDGARv4.2FT2012 | Total anthropogenic | 2000-2012 (yearly) | | EDGARv4.2FT2012 (2014), Olivier and Janssens-Maenhout (2014), Rogelj et al. (2014) |
| EDGARv4.2EXT | Fossil fuels, Agriculture and waste, biofuel | 1990-2013 (yearly) | | Based on EDGARv4.1 (EDGARv4.1, 2010), this study |
| USEPA | Fossil fuels, Agriculture and waste, biofuel, | 1990-2030 (10-yr interval, interpolated in this study) | | USEPA (2006, 2011, 2012) |
| IIASA GAINS ECLIPSE | Fossil fuels, Agriculture and waste, biofuel | 1990-2050 (5-yr interval, interpolated in this study) | X | Höglund-Isaksson (2012), Klimont et al. (2016) |
| FAOSTAT | Agriculture, Biomass Burning | Agriculture: 1961-2012 Biomass Burning: 1990-2014 | | Tubiello et al. (2013; 2015) |
| GFEDv3 | Biomass burning | 1997-2011 | X | van der Werf et al. (2010) |
| GFEDv4s | Biomass burning | 1997-2014 | X | Giglio et al. (2013) |
| GFASv1.0 | Biomass burning | 2000-2013 | X | Kaiser et al. (2012) |
| FINNv1 | Biomass burning | 2003-2014 | X | Wiedinmyer et al. (2011) |
| CLM 4.5 | Natural wetlands | 2000-2012 | X | Riley et al. (2011), Xu et al. (2016) |
| CTEM | Natural wetlands | 2000-2012 | X | Melton and Arora (2016) |
| DLEM | Natural wetlands | 2000-2012 | X | Tian et al., (2010;2015) |
| JULES | Natural wetlands | 2000-2012 | X | Hayman et al. (2014) |
| LPJ-MPI | Natural wetlands | 2000-2012 | X | Kleinen et al. (2012) |
| LPJ-wsl | Natural wetlands | 2000-2012 | X | Hodson et al. (2011) |
| LPX-Bern | Natural wetlands | 2000-2012 | X | Spahni et al. (2011) |
| ORCHIDEE | Natural wetlands | 2000-2012 | X | Ringeval et al. (2011) |
| SDGVM | Natural wetlands | 2000-2012 | X | Woodward and Lomas (2004), Cao et al. (1996) |
| TRIPLEX-GHG | Natural wetlands | 2000-2012 | X | Zhu et al., (2014;2015) |
| VISIT | Natural wetlands | 2000-2012 | X | Ito and Inatomi (2012) |



**Table 3: Average methane emissions over 2002-2006 and 2008-2012 at the global, latitudinal, and regional scales in Tg CH$_4$ yr$^{-1}$, and differences between the periods 2008-2012 and 2002-2006 from the top-down and the bottom-up approaches. Uncertainties are reported as [min-max] range of reported studies. Differences of 1 Tg CH$_4$ yr$^{-1}$ in the totals can occur due to rounding errors. A minimum of 3 years was required to calculate the average value over the 5-year periods, and then the difference between the two**
5  **periods was calculated for each approach. This means that 5 inversions are used to produce these values**

| | T-D | | | B-U |
|---|---|---|---|---|
| Period | 2002-2006 | 2008-2012 | 2012-2008 minus 2002-2006 | 2012-2008 minus 2002-2006 |
| GLOBAL | 546 [530-563] | 570 [546-580] | 22 [16-32] | 21 [5-41] |
| LATITUDINAL | | | | |
| 90°S- 30°N | 349 [330-379] | 363 [344-391] | 18 [13-24] | 6 [-4-13] |
| 30°N-60°N | 175 [158-194] | 184 [164-203] | 4 [0-9] | 17 [6-30] |
| 60°N-90°N | 20 [14-24] | 22 [15-31] | 0 [-1-1] | 0 [-3-3] |
| REGIONAL | | | | |
| Cent. North America | 10 [3-15] | 11 [6-16] | 2 [0-5] | 0 [0-1] |
| Tropical South America | 79 [60-97] | 94 [72-118] | 9 [6-13] | -2 [-6-2] |
| Temp. South America | 17 [12-27] | 15 [12-19] | 0 [-1-1] | 0 [-1-0] |
| Northern Africa | 41 [36-52] | 41 [36-55] | 2 [0-5] | 2 [0-5] |
| Southern Africa | 44 [37-54] | 45 [36-59] | 0 [-3-3] | 1 [-2-4] |
| South East Asia | 69 [53-81] | 73 [59-86] | 5 [-6-10] | 1 [-3-4] |
| India | 39 [28-45] | 37 [26-47] | 0 [-1-1] | 2 [1-3] |
| Oceania | 10 [7-19] | 10 [7-14] | 0 [0-1] | 0 [-1-1] |
| Contiguous USA | 42 [37-48] | 42 [33-48] | 1 [-2-3] | 2 [-1-4] |
| Europe | 27 [21-35] | 29 [22-36] | 1 [-1-3] | -2 [-2--2] |
| Central Eurasia & Japan | 46 [38-50] | 48 [38-58] | 1 [-1-6] | 5 [2-6] |
| China | 53 [47-62] | 56 [41-73] | 4 [1-11] | 10 [2-20] |
| Boreal North America | 19 [13-27] | 21 [15-27] | 0 [-3-3] | 2 [0-5] |
| Russia | 39 [32-45] | 38 [30-44] | -1 [-3-0] | 0 [-4-3] |

**Table 4: Mean values of the emission change (in Tg CH$_4$ yr$^{-1}$) between 2002-2006 and 2008-2012 inferred from the top-down and bottom-up approaches for the five general categories.**

| | Top-down | Bottom-up |
|---|---|---|
| Wetlands | 6 [-4-16] | -1 [-8-7] |
| Agriculture and waste | 10 [7-12] | 10 [7-13] |
| Fossil fuels | 7 [-2-16] | 17 [11-25] |
| Biomass burning | -3 [-7-0] | -2 [-5-0] |
| Other natural | 2 [-2-7] | - |



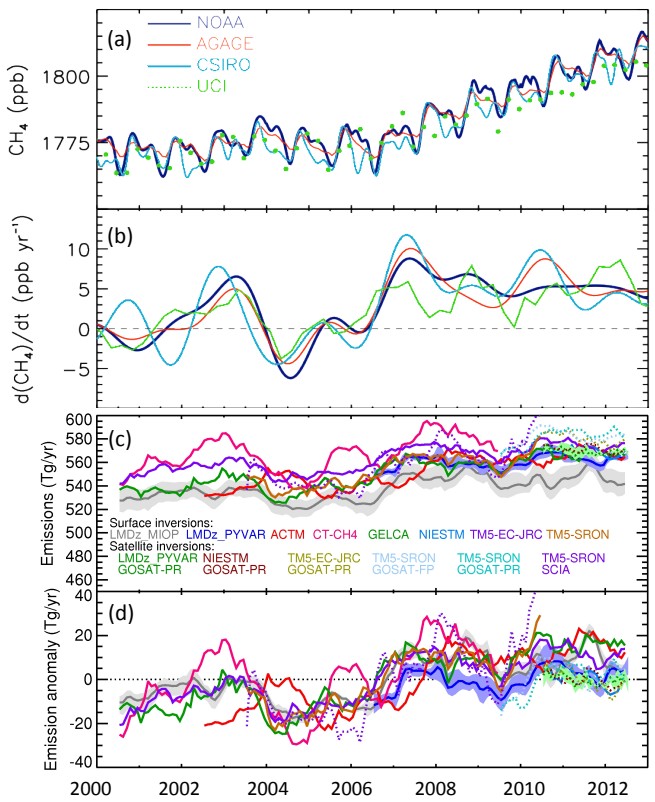

**Figure 1: Evolution of the global methane cycle since 2000. (a) Observed atmospheric mixing ratios (ppb) as synthetized for four different surface networks with a global coverage (NOAA, AGAGE, CSIRO, UCI). (b) Global Growth rate computed from (a) in**
5 **ppb/yr. 12-month running mean of (c) annual global emission (TgCH₄.yr⁻¹) and (d) annual global emission anomaly (TgCH₄.yr⁻¹) inferred by the ensemble of inversions.**



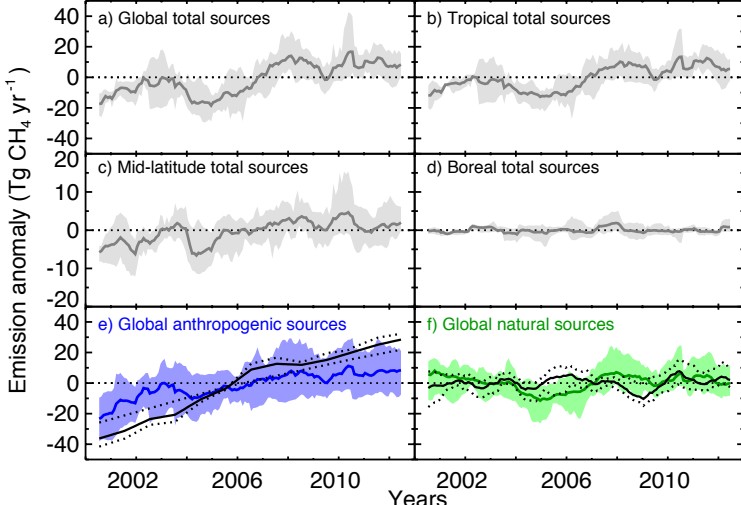

**Figure 2: 12-month running mean of annual methane emission anomalies (in Tg CH₄ yr⁻¹) inferred by the ensemble of inversions (mean as the solid line and min/max range as the shaded area) in grey for (a) global, (b) tropical, (c) mid-latitudes and (d) boreal total sources; in blue for (e) global anthropogenic sources and in green for (f) natural sources. The solid and dotted black lines** 5 **represent the mean and min/max range (respectively) of the bottom-up estimates: anthropogenic inventories in (e) and ensemble of wetland models in (f). The vertical scale is divided by 2 for the mid-latitude and boreal regions.**

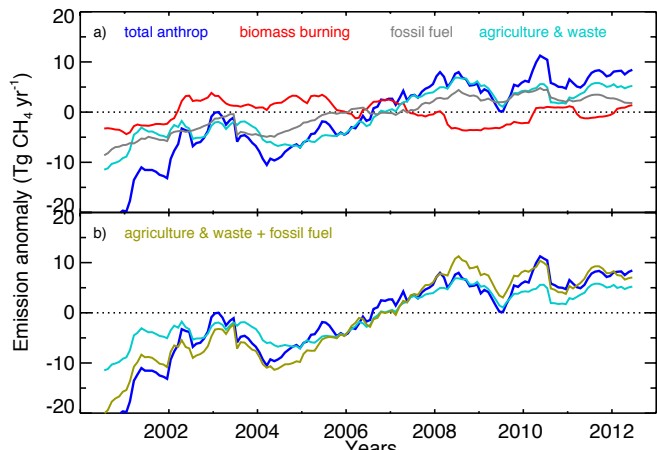

**Figure 3: 12-month running mean of global annual methane anthropogenic emission anomalies (Tg CH₄ yr⁻¹) inferred by the ensemble of inversions (only mean values of the ensemble are represented) for (a) total anthropogenic, biomass burning, fossil fuel** 10 **and, agriculture and waste sources. On the (b) panel, total anthropogenic and, agriculture and waste source anomalies are recalled on top of the sum of the anomalies from agriculture and waste, and fossil fuels sources.**





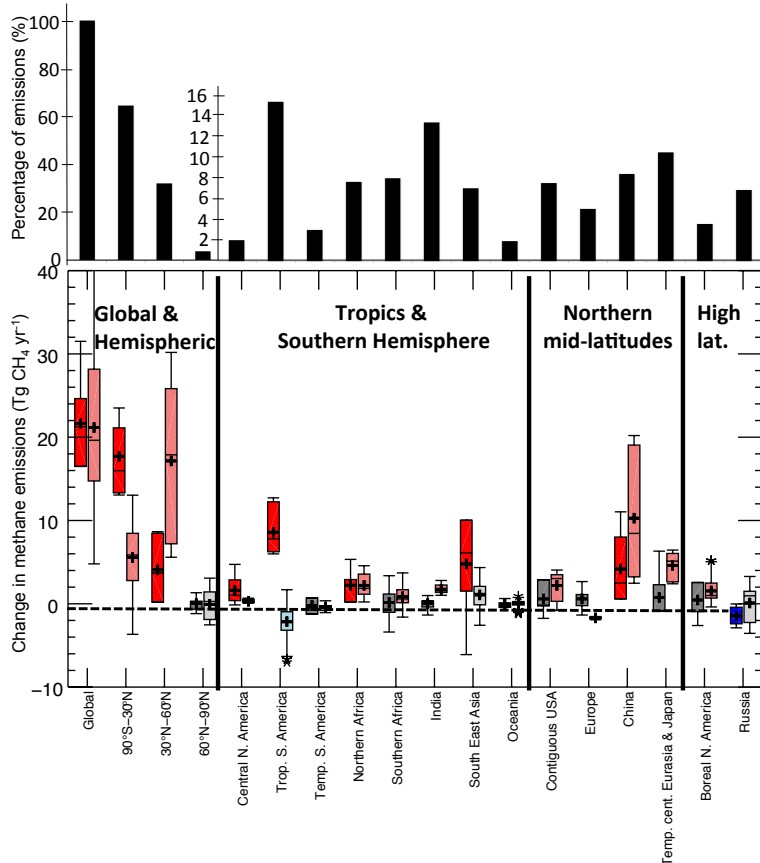

**Figure 4: Top: Contribution to the global methane emissions by region (in %, based on the mean top-down estimates over 2003-2012 from Saunois et al. (2016). Bottom: Changes in methane emissions 2002-2006 and 2008-2012 at global, hemispheric and regional scales in TgCH$_4$ yr$^{-1}$. Red boxplots indicate a significant positive contribution to emission changes (first and third quartiles above zero), blue boxplots indicate a significant negative contribution to emission changes (first and third quartiles below zero), grey boxplots indicate not-significant emission changes. Dark coloured boxes are for top-down (five long inversions) and light coloured for bottom-up approaches (see text for details). Median is indicated inside each boxplot (see Methods, section 2). Mean values, reported in the text, are represented with "+" symbols. Outliers are represented with stars. (Note: the bottom-up approaches that provide country estimates (and not maps, USEPA and FAOSTAT) have not been processed to provide hemispheric values. As a result the ensemble used for the three hemispheric regions differs from the ensemble used for the global and regional estimates. )**





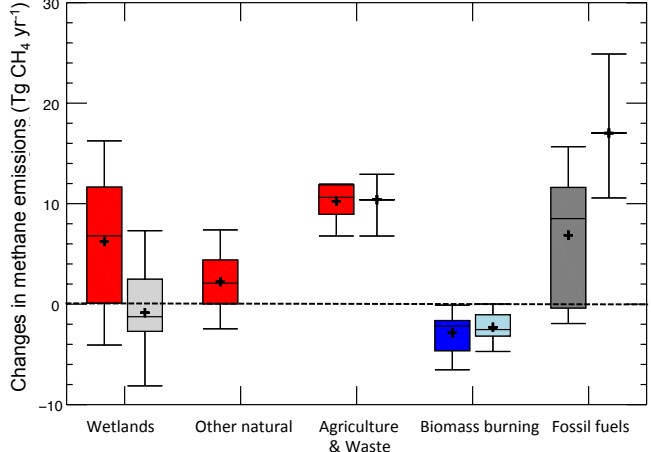

**Figure 5: Changes in methane emissions between 2002-2006 and 2008-2012 in Tg CH$_4$ yr$^{-1}$ for the five source types. Red boxplots indicate a significant positive contribution to emission changes (first and third quartiles above zero), blue boxplots indicate a significant negative contribution to emission changes (first and third quartiles below zero), grey boxplots indicate non-significant emission changes. Dark (light) coloured boxes are for top-down (bottom-up) approaches (see text for details). Median is indicated inside each boxplot (see Methods, Section 2). Mean values, reported in the text, are represented with "+" symbols.**



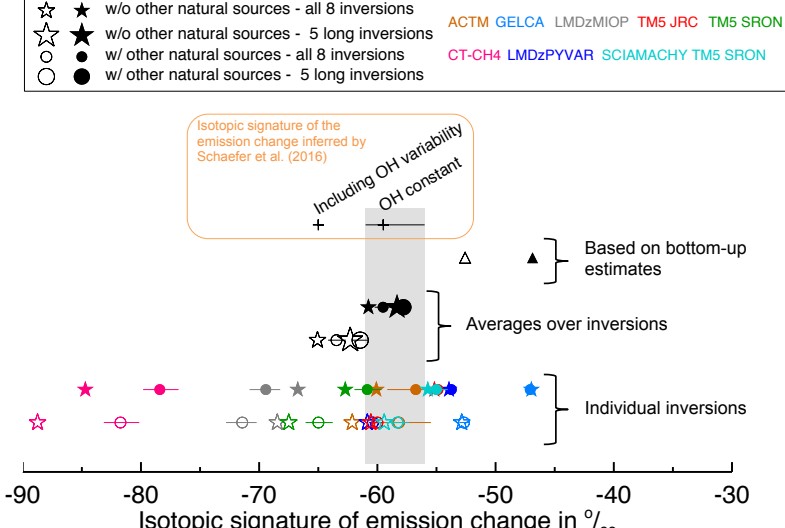

**Figure 6: Isotopic signature (in ‰) of the emission change between 2002-2006 and 2008-2012 based on Eq. 1 and the isotopic source signatures from Schaefer et al. (2016) and Schwietzke et al. (2016) in filled and open symbols respectively. The range of the isotopic signature of the emission change derived by the box-model of Schaefer et al. (2016) is indicated as the grey shaded area when assuming constant OH. The isotopic signatures derived from the ensemble of bottom-up estimates are shown with triangle symbol. The individual inversions are shown in colour. The mean inversion estimates are shown with stars and circles, without and with taking into account the "other natural" sources, respectively. The range around the circle indicates the range due to the choice of the isotopic source signature for the "other natural" source between -40 ‰ and -57 ‰ (see text).**