# Peer review of "Variability and quasi-decadal changes in the methane budget over the period 2000-2012"

_Atmospheric Chemistry and Physics, 2017_

## Referee Comment (RC1) · Anonymous Referee #1 · 17 May 2017

Comment on Saunois et al: Variability . . .in the methane budget

General This is a valuable update analysis of the GCP dataset used in the earlier 2016 paper by Saunois et al. This paper now focusses on understanding what is driving variability in methane mole fractions. This work is detailed and thorough, and is a valuable contribution to understanding what causes variability. The key 'missing factor' in the paper is a discussion of the impact of variability in the methane sinks. This gap is acknowledged, but could perhaps be discussed in more detail and paid more attention in qualifying the validity of the results. That said, the paper's implication that a step-change took place in 2006-8 (page 12, line 19) is very interesting and will need much future testing. A key factor not really discussed in much detail in this paper is

the time-response factor – how quickly do latitude-zonal methane mole fractions and particularly isotopes respond to a change in either sources or sinks?

Overall this paper is a valuable contribution and should be published with minor revision.

Specific Points Page 3 line 6 – models are not really 'data'. Line 9 – mention sinks? Line 24 – inconsistency with isotopes needs a little more highlighting here? Line 30 – This is the major weakness in the analysis and needs a bit more explanation. Page 4 Line 13 – maybe mention destruction of methane in caves/karsts, as it could be large? Line 19 – mention Rigby et al and Turner et al 2017? Line 34 – no trends from wetlands? – this is surely a very counter-intuitive finding given the enormous amount of water transferred onto the land in 2011, so much that the oceans fell (Boening et al, GRL, 39, L19602 ? Page 5 Line 14 – note Sherwood update of isotopic signature, 2017. Line 17 – agreed: ethane/methane is very uncertain and the source ratios may have changed greatly as the energy sources have changed. Line 21 – Rigby et al? Turner et al? Page 6 Line 13-14 – note that there is important seasonal, regional and latitudinal zonal information in the isotopes: cows - India; wetlands - SH S. America. Line 22 – OH – this is the missing elephant in the paper. Page 7 Line 10 – The problem with taking 2000-2012 is that the modelling may effectively seek to smooth over the really sharp year-on-year meteorological changes in the 2007-2011 period. Page 9 Line 18 – typo 'anomalies . . .shows' Page 10 Line 4 – note that gas use and coal use are heavily and variably dependent on meteorology – cold winter heating in China, or coal/gas fuelled electricity demand for air conditioning in the US and southern China in hot weather, etc etc. Line 19 – are cow populations in very cattle-rich Kenya, South Sudan, Cameroon, etc etc 'relatively stable?' – I doubt it. Are African cow populations increasing continuously? In Zimbabwe for example, cattle populations crashed in 2014. Page 11 Lines 1-10– all this assumes OH, soil sink, Cl destruction are not major factors. Line 16 – note the major reorganisations in the Chinese coal industry, and modernisation from many small gassy mines to fewer mines with better safety control

(methane). Line 19 – dry years in tropics Line 34- note Levin et als comment on Kai et al. Page 12 Line 3 – the problem of priors being EDGAR-dependent. . .could be discussed more? Line 13 – English problem – leads (?us) robustly to infer" Line 19 – key point of the whole paper. . . .step change. Page 13 Line 5 – note major coal industries in S Africa and Australia, and major Australian gas industry. Line 9 – lack of tropical observations – point also made by Bousquet et al some years ago – needs emphasis. Line 20 – choice of month – indeed. Page 14. Line 6 – N America: important point, needs emphasis. Line 14 – Arctic – another important point, needs emphasis. Line 18 – ditto. Page 15 Line 5 – this is very counter-intuitive given the flooding in Bolivia and the Amazon flows! Line 30 – maybe earlier estimates are over-dependent on production figures, and does not consider modernisation of mines. See also P 17 L7, which seems more realistic. Page 16 Line 4 – 2006-8 step change again. Line 15 – wetland variability near-zero?? Puzzling, given the la Nina 2011 flooding. – (also discussed on P 17 line 23: maybe it would be an idea to gather all this together?) Page 18 Line 32-33 – see new Sherwood inventory (ESSD 2017) Page 19 Line 14 – typo 20002 Line 20 – also better latitudinal information, especially in the tropics. Page 20 Line 6 – it is not clear that fracking in 2017 is now a major growth factor in emisions. Perhaps the opposite is happening. Various studies imply the frackers have really cut their gas losses in the past few years. Line 20 – 'even less changes' – clumsy English. Maybe rewrite whole sentence to make it clearer? Also Line 25 could be in clearer language, especially as it is an important sentence.

Conclusion

This is a valuable and interesting analysis of causes of variability. It does not properly address the sink problem, but nevertheless, once that gap is clearly acknowledged, it is a useful and significant contribution that should be published with minor revision.

---

## Referee Comment (RC2) · Anonymous Referee #2 · 2 Jun 2017

This is a timely, thoughtful, thorough, and important work form the scientific community involved in the Global Carbon Project. This effort focuses on the sub-decadal variability in an effort to the apparent, vexing shifts in the atmospheric growth rate of methane (CH4). It takes a measured approach to attributing the cause of CH4 variability in terms of natural (although perhaps perturbed by climate change) and direct anthropogenic sources. It also nicely consolidates the top-down inversions and the bottom-up emission inventories. It does not deal with the possible changes in atmospheric sinks (OH), although the evidence for large variability in the sink are proposed in some recent papers, but remain entirely obscure. This paper takes a balanced approach and could be published as is, or with some minor revisions suggested below. My apologies for the

delay in reading/reviewing this manuscript.

Request: Can we please all go to continuous line numbering so that it is easy to read sections and refer to them without finding the page number?

P4L18. The refs for OH chemistry models are fine as an overview, but the Holmes et al (2013, you have it later in the paper) is a good example of multi-model assessment of the interannual variability in the OH sink and its possible causes that is not found in the ACCMIP studies.

P5L15. The cryo-ethane history has proven useful in evaluating fossil fuel emissions and inferring ff-CH4 sources (Aydin et al., Nature 2011, 476:198-201, 2011; Nice-wonger, et al. GRL 43:214–221, 2015), and these are more relevant here than the LA-basin study of Wennberg.

P6L16-17. I think you mean "the first GCP global methane budget..."

P6L22ff. I think that this phrase is close but could be better "as most of the inversions used here assume constant OH concentrations over years, generally only optimizing its mean global concentration against methyl chloroform observations (e.g. Montzka et al. (2011))." What the models assume is not constant OH but rather constant CH4 loss frequency (with respect to OH). These are not the same, since if temperature changes then the constant OH will result in different CH4 loss. Moreover, the methyl chloroform decay records a mean loss frequency and not a mean OH as is frequently used. I suggest we move on to more accurate statements like: " as most of the inversions used here assume constant methane loss to OH over the time period, consistent with the observed decay of methyl chloroform (e.g. Montzka et al. (2011); Holmes et al, 2013)."

P9L11-15. You really need to note that if these models used their own OH & T fields that the CH4 budget would vary by 30% or more. It is because they use an accepted OH-lifetime for methane (e.g., Prather GRL 39:L09803, 2012) that top-down agrees so

closely.

P14L2. typo: constrained to constrain.

P14L8. While I tend to believe the Bruhwiler paper and not trust the cherry-picked satellite data over the US, you might consider referencing these two papers (Turner, Jacob, et al. GRL, 43:2218–2224, 2016; Schneising et al. Earth's Future 2:548–558, 2014).

P14L27. the phase "are assumed not to contribute" is awkward. At first it sound like this paper assumes this, but what you mean is "are assumed in these model studies not to .."?

P15L17. awkward end: "emissions occur partly over the same aereas...

P15L18. drop the 'of' to make it a sentence.

p15L23. Now you jump from fluxes (Tg/y) in the above to trends/accelerations (Tg/yr^2). How about using this line to transition to translate this difference in emissions to a trend: " and 2008-2012, i.e., a trend of about +1.7 Tg CH4 yr-2).

P16L2. ?? "change, and this result holds similarly for . . ."

P17L11. typo: "..that the increase in methane emissions between. . ."

P17L18ff. Please revise this sentence and make more, shorter ones. I was totally lost at the "although". " The sectorial partitioning from inversions is in agreement (within the uncertainty) with bottom-up inventories (noting that inversions are not independent from inventories), though the top-down ensemble significantly decreases the methane emission change from fossil fuel production and use compared to the bottom-up inventories, although the estimate of the latter should decrease with the upcoming revised version of the EDGAR inventory (see Sect. 3.2.4)."

P17L31. " the spread of land surface models" Please pick a better word than "spread": these models do not grow like forests. . ..

P18L8. Fix up: " wetland emissions per Meter Square." and put the Poulter ref at the end of the sentence if possible.

P18L16. I think you do not want 'incorrectly' in this sentence, the following clause says it all: "Even though top-down approaches may incorrectly attribute. . ."

P19L5. easier to read as: " ..changes leads, as expected, to unrealistically. . ."

P19L19. put a comma between the two independent clauses: " than constraints, and other. . ."

P19L21-35. Here is maybe where it is worth looking at the firn-air record showing ethane decreases (Aydin & Nicewonger refs above).

P20L3-6. This sentence does not really belong in the "Ethane" discussion? " Besides, the recent bottom-up study of Höglund-Isaksson (2017) shows relatively stable methane emissions from oil and gas after 2007….."

P20L17-29. This OH section is bothersome. I think you mean that models assume constant methane loss frequency – OR if they fix the 3D OH distribution, then the interannual temperature variations will drive changes in methane loss. I think they do the former and hence the correct wording would be "assume constant OH-lifetime for methane" or "assume constant methane loss frequency." These cannot just assume a uniform OH-loss because then they miss the seasonal and latitudinal gradients. I also recommend that the authors also look at the trends in methane's OH-lifetime from the Holmes et al 2013 paper. Several models show no trends from 2006 to 2010. If anything all the models show a decreasing methane OH-lifetime from a high in 2004 to a low in 2010, an 'OH' increase of about 3%. Moreover, one model running both with GEOS MERRA vs. GEOS6 shows different trends. The Dalsøren 2016 paper is very interesting, but it is only one model – further, this Oslo CTM3 shows different trends than the same model in the Holmes paper. I am not sure which is the better result, but some caution is due. Interestingly, all the models get the big increase in OH across the

1997-89 ENSO year.

P20L26. "However, decreasing OH concentrations since 2008 would require smaller emission changes to explain the observed atmospheric methane increase, also possibly implying .." This is confusing since both the Dalsoren and Holmes papers show a decrease in lifetime (2% possibly) and hence an increase in OH after 2008.

P21L27-30. Again, please check that the models kept the methane OH-lifetime (effectively the inverse loss frequency) fixed and did not freeze OH concentrations, allowing the rate coefficient to vary with temperature as it should, because then the temperature fluctuations could drive %-level variability. Also I think you have the Dalsoren paper backwards: their Fig 15 (&18) shows a steadily increasing methane loss frequency (1/lifetime, left scale) since the 1997-98 ENSO and up to 2010; the year 2008 is the only reversal of this. Their calculated change in OH does not match the CH4 lifetime, and it is the lifetime that determines the annual loss of CH4.

P21L33. "uncertainties" is odd. I am not sure we know enough to even assess the uncertainty. how about "major disagreements in OH fields simulated by the models."

P22L1. It is the fact that we stopped using MCF and it is decreasing rapidly, that makes is a good surrogate for the methane OH-lifetime. When in use the uncertainty in emissions made it difficult to get better than 10-20% accuracy and variability.

P22L5. I am not sure that this comparison with CO2 is useful or accurate. There are many thorny problems left with the CO2 budget and climate feedbacks. Stop at "understood."

---

## Author Comment (AC1) · 18 Jul 2017

**Detailed Response to Anonymous Referee #1**

We acknowledge anonymous referee #1 for his/her time spent on reading and commenting on the paper, providing comments and helpful suggestions to improve the manuscript.

*General*
*This is a valuable update analysis of the GCP dataset used in the earlier 2016 paper by Saunois et al. This paper now focusses on understanding what is driving variability in methane mole fractions. This work is detailed and thorough, and is a valuable contribution to understanding what causes variability. The key 'missing factor' in the paper is a discussion of the impact of variability in the methane sinks. This gap is acknowledged, but could perhaps be discussed in more detail and paid more attention in qualifying the validity of the results. That said, the paper's implication that a step-change took place in 2006-8 (page 12, line 19) is very interesting and will need much future testing. A key factor not really discussed in much detail in this paper is the time-response factor – how quickly do latitude-zonal methane mole fractions and particularly isotopes respond to a change in either sources or sinks? Overall this paper is a valuable contribution and should be published with minor revision.*

Although not addressing fully the contribution of OH to methane changes from 2000 to 2012, we extended the discussion in the text.
A more detailed discussion on the "time-response factor" mentioned by the reviewer would necessitate specific simulations from sector with various tracers and would be more a TRANSCOM-like experiment (model inter-comparison project) and is beyond the scope and objectives of this review on existing simulations and inversions.

*Specific Points*

*Page 3 line 6 – models are not really 'data'.*

The sentence has been rephrased as follow: "**The GCP dataset integrates results from top-down studies (exploiting atmospheric observations within an atmospheric inverse-modelling frameworks) and bottom-up models (including process-based models for estimating land surface emissions and atmospheric chemistry), inventories of anthropogenic emissions, and data-driven approaches.**"

*Line 9 – mention sinks?*

The sink variability and trends are not fully discussed in the paper. A statement on the sink changes is provided at the end of the abstract and a paragraph has been further developed at the end of the paper before conclusion. However, we do not pretend in the paper to fully address the OH related question as stated early in the text.

*Line 24 – inconsistency with isotopes needs a little more highlighting here?*

The sentence has been modified in order be more clear to the reader without having to read the whole text.

"**We apply isotopic signatures to the emission changes estimated for individual studies based on five emission sectors and find that for six individual top-down studies (out of eight) the average isotopic signature of the emission changes is not consistent with the observed change in atmospheric $^{13}CH_4$. However the partitioning in emission change derived from the ensemble mean is consistent with this isotopic constraint.**"

*Line 30 –This is the major weakness in the analysis and needs a bit more explanation*

This sentence has been re-written has follow to explain why OH sink has not been studied here.

**"In most of the top-down studies included here, OH concentrations are considered constant over the years (seasonal variations but without any inter annual variability). As a result, the methane loss (in particular through OH oxidation) varies mainly through the change in methane concentrations and not its oxidants. For these reasons, changes in the methane loss could not be properly investigated in this study, although it may play a significant role in the recent atmospheric methane changes as briefly discussed at the end of the paper."**

*Page 4*

*Line 13 – maybe mention destruction of methane in caves/karsts, as it could be large?*

Subterranean methane sinks have been studied at local scale and show, indeed, that these sinks can be large on the local scale. However the impact of the karst sink on the global methane budget is unknown. Besides, the processes involved are far from being totally understood, and, unfortunately, assessed at the global scale. The karst/caves methane sink has not been mentioned in the GCP methane budget ESSD paper, however we keep this (mostly) open question in mind and look forward hearing about regional/global assessment of this sink, so that it could be eventually included in the next methane budget.

*Line 19 – mention Rigby et al and Turner et al 2017?*

At this line, the list of articles refers to OH derived from climate models. The suggested references do not match this list. However these two recent papers need to be cited in this review, which is done former page 5, line 25, as suggested below.

*Line 34 – no trends from wetlands?– this is surely a very counter-intuitive finding given the enormous amount of water transferred onto the land in 2011, so much that the oceans fell (Boening et al, GRL, 39, L19602 ?*

Here the trend over 2000-2012 is discussed. This does not mean that wetland emissions have not experienced large year-to-year variations in particular in 2010-2011. Indeed the large year-to-year variations in methane emissions from wetlands make it difficult to find significant trend. Also the statement here is on the global trend in wetland emissions; as discussed later, the quasi "null" trend in global emissions results in increasing wetland emissions in the mid and high latitudes counter balanced by decreasing emissions in the Tropics, over the aforementioned full 13 year-period. We added the sentence: **"This flat trend over the decade is associated to large year-to-year variations (e.g. 2010-11 in the tropics) that limits its robustness together with sensitivities to the choice of the inventory chosen to represent the wetland extend".**

*Page 5 Line 14 – note Sherwood update of isotopic signature, 2017.*

The Sherwood et al., 2017 review of isotopic signature has been added and cited along with Schwietzke et al., 2016 study, as follows, former page 5, line 8: "**Schwietzke et al. (2016), using updated estimates of the source isotopic signatures (Sherwood et al., 2017) with rather narrow uncertainty ranges..**"

*Line 17 – agreed: ethane/methane is very uncertain and the source ratios may have changed greatly as the energy sources have changed.*

Comment that does not require specific answer.

*Line 21 – Rigby et al? Turner et al?*

These two recent references (not published at the moment of submission) have been added at the end of the paragraph (because they are more recent and need some details (as follows). Rigby et al. has also been added on former page 6 line 25.

**"[...] Dalsoren et al. (2016) found constant OH concentrations since 2007, and Rigby et al. (2017) a decrease in OH concentrations, both results possibly contributing to the**

**observed increase in methane growth rate and therefore limiting the required changes in methane emissions inferred by top-down studies. However Turner et al. (2017) highlight the difficulty in disentangling the contribution in emission or sink changes when OH concentrations are weakly constrained by atmospheric measurements."**

*Page 6 Line 13-14 – note that there is important seasonal, regional and latitudinal zonal information in the isotopes: cows - India; wetlands - SH S. America.*

Yes, there are different regional and seasonal variations in the emissions that help interpreting the methane signal and its isotopic signal. We completed the sentence in the text: … **"or to separate regions with a dominant source (e.g. agriculture in India versus wetlands in Amazonia),"**

*Line 22 – OH – this is the missing elephant in the paper.*

As already answered before, we acknowledge the weakness of this review regarding the lack of discussion on OH changes. Unfortunately, today, atmospheric inversion studies struggle dealing with OH variability over the years. In the present review, the results presented here should be taken as "what would be the emission changes considering constant OH over the years". However, we added few sentences of discussion at the end of the text before the conclusions. For the future this issue should be addressed.

*Page 7 Line 10 – The problem with taking 2000-2012 is that the modelling may effectively seek to smooth over the really sharp year-on-year meteorological changes in the 2007-2011 period.*

The choice of the 2000-2012 period responds to the willingness of better understanding the recent changes in atmospheric methane and corresponds to a period of large amount of atmospheric methane observations. Regarding the sharp meteorological year to year changes in the 2007-2011 period, they are accounted for in the atmospheric transport and the models do show some important year to year variations on methane emissions, especially around 2010-2011, as shown for example on Fig 1c (it should be noted that 12 month running means are displayed, smoothing a bit the monthly variations). So we do not see major reasons that the models especially miss some variability between 2007 and 2011.

*Page 9 Line 18 – typo 'anomalies : : :shows'*

Thank you for pointing the typo. This has been corrected

*Page 10 Line 4 – note that gas use and coal use are heavily and variably dependent on meteorology – cold winter heating in China, or coal/gas fuelled electricity demand for air conditioning in the US and southern China in hot weather, etc etc.*

Indeed we missed the potential climate variability of fuel demand. The former sentence, page 10, line 7 has ben modified as follows: "**Fossil-fuel exploitation can also be sensitive to rapid economic changes, and meteorological variability may impact the fuel demand for heating and cooling systems.**"

*Line 19 – are cow populations in very cattle-rich Kenya, South Sudan, Cameroon, etc etc 'relatively stable?' – I doubt it. Are African cow populations increasing continuously? In Zimbabwe for example, cattle populations crashed in 2014.*

At the continental scale of Africa, the statistics data show continuously increasing cattle population. However, country specific study of FAO statistics would probably show more year-to-year changes in cattle and buffaloes: for example Kenya experienced a rapid increase in cattle population between 2006 and 2008, though Kenya represents only 5% of the African cattle population. Looking further into FAO statistics per country, one may see that some statistics are missing for some African countries for the earlier years (South Sudan). Regarding Zimbabwe, it should be acknowledge that Zimbabwe represents less than 2% of the African cattle population and that the recent changes (after 2012) are not included in our study.

*Page 11 Lines 1-10– all this assumes OH, soil sink, Cl destruction are not major factors.*

For most of the models, the soil sink is from climatological estimates. As a result, the entire discussion in the paper is on emission changes assuming constant OH and Cl concentrations, and soil sink. The chemical sink, per se, is not constant as it depends on methane concentrations. In the introduction (formerly page 6, from line 21), we modified our sentences to be clearer from the beginning on this issue and we reinforced the last paragraph of the paper as well.

**"However, we do not address the contribution of the methane sinks during this period. Indeed, for most of the models, the soil sink is from climatological estimates and the oxidant concentration fields (OH, Cl, O1D) are assumed constant over the years. The global mean of OH concentrations was generally optimized against methyl chloroform observations (e.g. Montzka et al. (2011)), but no inter annual variability is applied."**

*Line 16 – note the major reorganisations in the Chinese coal industry, and modernisation from many small gassy mines to fewer mines with better safety control (methane).*

The following sentence has been added to acknowledge this, however including such considerations on modernization of coal exploitation in China will probably not counterbalance the large increase in Chinese coal production.

**"This recent period is characterized by major re-organizations in the Chinese coal industry, including evolution from many small gassy mines to fewer mines with better safety and emission control."**

*Line 19 – dry years in tropics*

This explanation has been added as follows:

**"However, between 2002 and 2010, a significant negative trend of -0.5±0.1 Tg CH4 yr-2 is found for biomass burning, both from the top-down approaches (Fig. S5) and the GFED3 and GFED4s inventory (Fig. S10), this corresponds to dry years in the tropics. Although it should be noted that almost all inversions use GFED3 in their prior (Table S1) and therefore are not independent from the bottom-up estimates."**

*Line 34- note Levin et al comment on Kai et al.*

We have added the following sentence to acknowledge the dependency on the data selection made by Kai et al.

**"However Levin et al. (2012) showed that the isotopic data selection might bias this result, as they found no such decrease when using background site measurements."**

*Page 12 Line 3 – the problem of priors being EDGAR-dependent: : :could be discussed more?*

We rephrased this sentence: **"However, the estimated anthropogenic emissions can significantly deviate from this common prior. Similarly, inversions based on the same prior wetland fluxes do not systematically infer the same variations in methane total and natural emissions. These different increments from the prior are constrained by atmospheric observations and qualitatively indicate that inversions can depart from prior estimates."**

*Line 13 – English problem – leads (?us) robustly to infer"*

This has been changed to: "**Even using time-constant prior emissions for fossil fuels in the inversions results in robustly inferring increasing fossil fuel emissions [...]**"

*Line 19 – key point of the whole paper: : :.step change.*

This step change in the inversion is discussed in the discussion section in the "Methane sink by OH" part. We have added the following sentence on former page 12 line 20:

**"The requirement of a step change in the emissions will be further discussed in Section 4."**

*Page 13 Line 5 – note major coal industries in S Africa and Australia, and major Australian gas industry.*

It seems hard to enter in such a precision in this part where we present broader results.

*Line 9 – lack of tropical observations – point also made by Bousquet et al some years ago – needs emphasis.*

The reference has been added here as follows: **"Yet most of inversions rely on surface observations, which poorly represent the tropical continents, as previously noticed by previous individual study (e.g., Bousquet et al. (2011)"**

*Line 20 – choice of month – indeed.*

We rephrased: **"... sensitive to the choice of the starting and ending dates of the time period"**

*Page 14. Line 6 – N America: important point, needs emphasis.*

We have added the previous Turner et al (2016?) paper reference and update the Bruhwiler et al reference that have been published since the submission. The paragraph as been changed as follows: **"Also, temperate North America does not contribute significantly to the emission changes. Contrary to a large increase in the US emissions suggested by Turner et al. (2016), none of the inversions detect, at least prior to 2013, an increase in methane emissions possible due to increasing shale gas exploitation in the U.S. Bruhwiler et al. (2017) highlight the difficulty of deriving trends on relatively short term due to in particular inter annual variability in transport."**

*Line 14 – Arctic – another important point, needs emphasis.*

*Line 18 – ditto.*

We rephrased and extended the already existing paragraph about the Arctic: p14 l10-26, as follows:

**"Permafrost thawing may have caused additional methane production underground (Christensen et al., 20014) but changes in the out coming methane flux to the atmosphere, possibly hidden in wetland emissions under existing wetlands, has not been detected by continuous atmospheric stations around the Arctic, despite a small increase in late autumn/early winter in methane emission from Arctic tundra, (Sweeney et al., 2016). However, unintentional double counting of emissions from different water systems (wetlands, rivers, lakes) may lead to Artic emission growth in the bottom-up studies when little or none exists (Thornton et al., 2016). The detectability of possibly increasing methane emissions from the Arctic seems possible today based on the continuous monitoring of Arctic atmosphere at few but key stations (e.g., Berchet et al., 2016; Thonat et al., 2017), but this surface network remains fragile on the long-term and would be more robust with additional constraints such as those that will be provided in 2021 by the active satellite mission MERLIN (Pierangello et al., 2016; Kiemle et al., 2014)."**

*Page 15 Line 5 – this is very counter-intuitive given the flooding in Bolivia and the Amazon flows!*

We agree but this result is mostly driven by the inventory for wetland extent used by all bottom-up models (see Poulter et al., 2017). We recall this in the text:

**"[...] mostly due to a reduction in tropical wetland extent, as constrained by the common inventory used by all models, see Poulter et al., 2017)"**

*Line 30 – maybe earlier estimates are over-dependent on production figures, and does not consider modernisation of mines. See also P 17 L7, which seems more realistic.*

We already added this remark in page 11 line 16 comment.

*Page 16 Line 4 – 2006-8 step change again.*

As discussed previously, the "step-change" might be emphasized by the assumption of constant OH concentrations in the inversions. This is discussed in the discussion section. As we acknowledge this weakness, we decided to not define it as a "step change".

*Line 15 – wetland variability near-zero?? Puzzling, given the la Nina 2011 flooding. – (also discussed on P 17 line 23: maybe it would be an idea to gather all this together?)*

Here the "emission change" between the two period is discussed not the "inter annual variability". There could be no change between the two periods, but inter annual variability within each period, which is the case. And this is why it is hard to derive trend from a signal with large inter annual variability. Due to the importance of the source and its variation, we chose to further discuss it the Discussion Section 4.

*Page 18 Line 32-33 – see new Sherwood inventory (ESSD 2017)*

Sherwood et al. 2017 as been added as follows on former page 19, line 1: **"[...] while a recent study suggests different globally averaged isotopic signatures (Sherwood et al., 2017), with a lighter fossil fuel..."**

*Page 19 Line 14 – typo 20002*

This has been corrected

*Line 20 – also better latitudinal information, especially in the tropics.*

We modified the sentence: **"This problem has more unknowns than constraints, and other pieces of information need to be added to further solve it (such as $^{14}$C, deuterium, or co-emitted species but also better latitudinal information, especially in the tropics)."**

*Page 20 Line 6 – it is not clear that fracking in 2017 is now a major growth factor in emissions. Perhaps the opposite is happening. Various studies imply the frackers have really cut their gas losses in the past few years.*

Our study ends in 2012. The recent change in fracking cannot be addressed here but might be – if any signal appears, in the next GCP exercise.

*Line 20 – 'even less changes' – clumsy English. Maybe rewrite whole sentence to make it clearer? Also Line 25 could be in clearer language, especially as it is an important sentence.*

This paragraph about methane sinks has been rephrased and extended to include recent papers, as stated several times before.

*Conclusion*

*This is a valuable and interesting analysis of causes of variability. It does not properly address the sink problem, but nevertheless, once that gap is clearly acknowledged, it is a useful and significant contribution that should be published with minor revision.*

---

## Author Comment (AC2) · 18 Jul 2017

**Detailed Response to Anonymous Referee #2**

We acknowledge anonymous referee #2 for his/her time spent on reading and commenting on the paper, providing comments and helpful suggestions to improve the manuscript, in particular the English and phrasing of some sentences, and citations that were missing.

*This is a timely, thoughtful, thorough, and important work form the scientific community involved in the Global Carbon Project. This effort focuses on the sub-decadal variability in an effort to the apparent, vexing shifts in the atmospheric growth rate of methane (CH4). It takes a measured approach to attributing the cause of CH4 variability in terms of natural (although perhaps perturbed by climate change) and direct anthropogenic sources. It also nicely consolidates the top-down inversions and the bottom-up emission inventories. It does not deal with the possible changes in atmospheric sinks (OH), although the evidence for large variability in the sink are proposed in some recent papers, but remain entirely obscure. This paper takes a balanced approach and could be published as is, or with some minor revisions suggested below. My apologies for the delay in reading/reviewing this manuscript. Request: Can we please all go to continuous line numbering so that it is easy to read sections and refer to them without finding the page number?*

*P4L18. The refs for OH chemistry models are fine as an overview, but the Holmes et al (2013, you have it later in the paper) is a good example of multi-model assessment of the interannual variability in the OH sink and its possible causes that is not found in the ACCMIP studies.*

We thank the reviewer for the additional **reference** that has been **added** at this place of the text.

*P5L15. The cryo-ethane history has proven useful in evaluating fossil fuel emissions and inferring ff-CH4 sources (Aydin et al., Nature 2011, 476:198-201, 2011; Nicewonger, et al. GRL 43:214–221, 2015), and these are more relevant here than the LA-basin study of Wennberg.*

We thank the reviewer for these two relevant papers, both using cryo-observations of the atmospheric composition. **These two papers** have been **added** on top of the Wennberg et al., which used in-situ present atmospheric observations.

*P6L16-17. I think you mean "the first GCP global methane budget :"*

Indeed, it would not be fair to pretend that other methane reviews do not exist. This has been modified.

*P6L22ff. I think that this phrase is close but could be better "as most of the inversions used here assume constant OH concentrations over years, generally only optimizing its mean global concentration against methyl chloroform observations (e.g. Montzka et al. (2011))." What the models assume is not constant OH but rather constant CH4 loss frequency (with respect to OH). These are not the same, since if temperature changes then the constant OH will result in different CH4 loss. Moreover, the methyl chloroform decay records a mean loss frequency and not a mean OH as is frequently used. I suggest we move on to more accurate statements like: " as most of the inversions used here assume constant methane loss to OH over the time period, consistent with the observed decay of methyl chloroform (e.g. Montzka et al. (2011); Holmes et al, 2013)."*

We thank the reviewer for this very interesting comment. Actually, there is probably a misunderstanding in the way MCF and CH4 inversions are done. In the chemistry transport models used in inverse modeling, the chemical loss of a compound through OH is calculated at each time step using OH (prescribed), the compound concentrations and the reaction constant (driven by the temperature 3D field generally nudged to ECMWF inter-annual reanalyses). For MCF inversions a scaling factor is optimized for the loss, but then it is used to get inter annually varying OH (few inversion) or a seasonnaly-varing climatological OH

(most inversions), that is then prescribed to CH4 inversions. Therefore, CH4 inversions do not prescribe the pre-optimized loss but the MCF-derived OH fields, which may introduce some inconsistency about the impact of temperature changes on the loss. This effect is absorbed in OH variations as it is done currently. It remains probably small when looking only to 1-2 decades but could be significant during large climate events such as El Niño.

This part has been re-written as follows: " **However, we do not address the contribution of the methane sinks during this period. Indeed, for most of the models, the soil sink is from climatological estimates and the oxidant concentration fields (OH, Cl, O1D) are assumed constant over the years. The global mean of OH concentrations was generally optimized against methyl chloroform observations (e.g. Montzka et al. (2011)), but no inter annual variability is applied.**"

*P9L11-15. You really need to note that if these models used their own OH & T fields that the CH4 budget would vary by 30% or more. It is because they use an accepted OH-lifetime for methane (e.g., Prather GRL 39:L09803, 2012) that top-down agrees so closely.*

This is true that the agreement between the inversions is linked to the use of similar OH concentrations and temperature fields. We have added the following sentence: "**It is to be noted that this rather good agreement between these estimates is linked with the associated rather small range of global sinks. Indeed, most inversions use similar MCF-constrained OH fields and temperature fields.**"

*P14L2. typo: constrained to constrain.*

This has been corrected

*P14L8. While I tend to believe the Bruhwiler paper and not trust the cherry-picked satellite data over the US, you might consider referencing these two papers (Turner, Jacob, et al. GRL, 43:2218–2224, 2016; Schneising et al. Earth's Future 2:548–558, 2014).*

Following your comment and reviewer#1 comment on this paragraph. The paragraph as been changed as follows: "**Also, temperate North America does not contribute significantly to the emission changes. Contrary to a large increase in the US emissions suggested by Turner et al. (2016), none of the inversions detect, at least prior to 2013, an increase in methane emissions possible due to increasing shale gas exploitation in the U.S. Bruhwiler et al. (2017) highlight the difficulty of deriving trends on relatively short term due to in particular inter annual variability in transport.**"

*P14L27. the phrase "are assumed not to contribute" is awkward. At first it sound like this paper assumes this, but what you mean is "are assumed in these model studies not to .."?*

This has been rephrased as suggested.

*P15L17. awkward end: "emissions occur partly over the same areas:*

This has been rephrased to: "**emissions may both occur in the same or neighboring model pixels.**"

*P15L18. drop the 'of' to make it a sentence.*

This has been corrected.

*p15L23. Now you jump from fluxes (Tg/y) in the above to trends/accelerations (Tg/yr^2). How about using this line to transition to translate this difference in emissions to a trend: " and 2008-2012, i.e., a trend of about +1.7 Tg CH4 yr-2).*

Thank you for pointing this. The suggested change has been done as follows: "**For China, bottom-up approaches suggest a +10 [2-20] Tg $CH_4$ $yr^{-1}$ emission increase between 2002-2006 and 2008-2012, i.e. a trend of about 1.7 Tg $CH_4$ $yr^{-2}$ (considering a 10 Tg $yr^{-1}$ increase over 2004-2010), which is much larger than the top-down estimates.**"

*P16L2. ?? "change, and this result holds similarly for : : :"*

Thank you for the rewriting, this has been modified.

*P17L11. typo: "..that the increase in methane emissions between: : :"*

This has been corrected.

*P17L18ff. Please revise this sentence and make more, shorter ones. I was totally lost at the "although". " The sectorial partitioning from inversions is in agreement (within the uncertainty) with bottom-up inventories (noting that inversions are not independent from inventories), though the top-down ensemble significantly decreases the methane emission change from fossil fuel production and use compared to the bottom-up inventories, although the estimate of the latter should decrease with the upcoming revised version of the EDGAR inventory (see Sect. 3.2.4)."*

Indeed... This has been changed to: **"The sectorial partitioning from inversions is in agreement (within the uncertainty) with bottom-up inventories (noting that inversions are not independent from inventories). However the top-down ensemble significantly decreases the methane emission change from fossil fuel production and use compared to the bottom-up inventories. In the coming years, the revised version of the EDGAR inventory (see Sect. 3.2.4) should decrease the estimated change by bottom-up inventories, reducing the difference between bottom-up and top-down estimates."**

*P17L31. " the spread of land surface models" Please pick a better word than "spread": these models do not grow like forests:*

This has been rephrased to: **"The range of the methane emissions estimated by land surface models driven with the same flooded area extent shows that [...]"**

*P18L8. Fix up: " wetland emissions per Meter Square." and put the Poulter ref at the end of the sentence if possible.*

The sentence has been rephrased as follows: **"However, no significant trend in tropical surface temperature is inferred over 2000-2012 that could explain an increase in tropical wetland emissions (Poulter et al., in review)."**

*P18L16. I think you do not want 'incorrectly' in this sentence, the following clause says it all: "Even though top-down approaches may incorrectly attribute: : :"*

'incorrectly' has been removed from the sentence as follows: **"Even though top-down approaches may attribute the emissions increase between 2002-2006 and 2008-2012 to tropical regions (and hence partly to wetland emitting areas) due to a lack of observational constraints, it is not possible, with the evidence provided in this study, to rule out a potential positive contribution of wetland emissions in the increase of global methane emissions at the global scale."**

*P19L5. easier to read as: " ..changes leads, as expected, to unrealistically: : :"*

This has been corrected accordingly.

*P19L19. put a comma between the two independent clauses: " than constraints, and other :"*

This has been added in the sentence.

*P19L21-35. Here is maybe where it is worth looking at the firn-air record showing ethane decreases (Aydin & Nicewonger refs above).*

Nicewonger et al. results span only to 1918 and could not provide any insight in this discussion on the recent change. The Aydin study used firn air data and discussed fossil fuel emissions change from 1900 to 2010. This is not completely compatible with the period discussed here so we added a sentence about these historical papers in the introduction:

**"The historical record of atmospheric ethane suggests an increase of ethane sources until the 1980s and then a decrease driven by fossil fuel related emissions until the early 2000s (Aydin et al., 2011)."**

*P20L3-6. This sentence does not really belong in the "Ethane" discussion? " Besides, the recent bottom-up study of Höglund-Isaksson (2017) shows relatively stable methane emissions from oil and gas after 2007: : :."*

The Höglund-Isaksson study does not use ethane measurement, however they show constant emission from the oil and gas sector. This result disagrees with the ethane-based study and is worth noting in this context. As a result, we decided to change the paragraph title to **"Oil and gas emissions, and ethane constraint"**

*P20L17-29. This OH section is bothersome. I think you mean that models assume constant methane loss frequency – OR if they fix the 3D OH distribution, then the interannual temperature variations will drive changes in methane loss. I think they do the former and hence the correct wording would be "assume constant OH-lifetime for methane" or "assume constant methane loss frequency." These cannot just assume a uniform OH-loss because then they miss the seasonal and latitudinal gradients.*

As explained above this is the second statement that occurs in practice in current inversions. Indeed, inverse modellers prescribed climatological OH (with seasonal variations) and compute the loss using varying CH4 and temperature. This might not be fully consistent but we clarified the method in the text.

*I also recommend that the authors also look at the trends in methane's OH-lifetime from the Holmes et al 2013 paper. Several models show no trends from 2006 to 2010. If anything all the models show a decreasing methane OH-lifetime from a high in 2004 to a low in 2010, an 'OH' increase of about 3%. Moreover, one model running both with GEOS MERRA vs. GEOS6 shows different trends. The Dalsøren 2016 paper is very interesting, but it is only one model – further, this Oslo CTM3 shows different trends than the same model in the Holmes paper. I am not sure which is the better result, but some caution is due. Interestingly, all the models get the big increase in OH across the 1997-89 ENSO year.*

We acknowledge the caution suggested by the reviewer. **And modified the text accordingly in the last paragraph of the paper.**

*P20L26. "However, decreasing OH concentrations since 2008 would require smaller emission changes to explain the observed atmospheric methane increase, also possibly implying .." This is confusing since both the Dalsoren and Holmes papers show a decrease in lifetime (2% possibly) and hence an increase in OH after 2008.*

Figure 1 of Holmes 2013 and figure 15 of Dalsoren 2016 are consistent until 2007. Holmes stops in 2009 but Dalsoren shows stabilizing OH after 2007. **We rephrased this paragraph to better show the remaining uncertainties on OH variations.**

*P21L27-30. Again, please check that the models kept the methane OH-lifetime (effectively the inverse loss frequency) fixed and did not freeze OH concentrations, allowing the rate coefficient to vary with temperature as it should, because then the temperature fluctuations could drive %-level variability. Also I think you have the Dalsoren paper backwards: their Fig 15 (&18) shows a steadily increasing methane loss frequency (1/lifetime, left scale) since the 1997-98 ENSO and up to 2010; the year 2008 is the only reversal of this. Their calculated change in OH does not match the CH4 lifetime, and it is the lifetime that determines the annual loss of CH4.*

Again, as explained above, CTMs implied in inversions prescribe OH change and recomputed the loss using inter-annually varying meteorology. T**he paragraph was rephrased to better reflect what was done and the remaining uncertainties**.

*P21L33. "uncertainties" is odd. I am not sure we know enough to even assess the uncertainty. how about "major disagreements in OH fields simulated by the models."*

We agree on this comment and have changed the sentence as suggested to: "**Estimating and optimizing OH oxidation in top-down approaches is challenging due to the major disagreements in OH fields simulated by the models.**"

*P22L1. It is the fact that we stopped using MCF and it is decreasing rapidly, that makes is a good surrogate for the methane OH-lifetime. When in use the uncertainty in emissions made it difficult to get better than 10-20% accuracy and variability.*

We agree and have rephrased this part of the conclusion: **"Although beneficial for the recovery of the stratospheric ozone, methyl-chloroform, which is used as a proxy to derive OH variations, is decreasing rapidly in the atmosphere. MCF is therefore less sensitive to uncertain and larger emissions as in the 1980s and 1990s (e.g. Kroll et al., 2003; Prinn et al., 2001), but within years, will also be less useful to derive OH changes as its atmospheric concentrations are getting as small as the precision and accuracy of the measurements. "**

*P22L5. I am not sure that this comparison with CO2 is useful or accurate. There are many thorny problems left with the CO2 budget and climate feedbacks. Stop at "understood."*

The sentence has been stopped at "understood" as suggested.